# PudgyTurtle Mode Resists Bit-Flipping Attacks

David A. August [1,*] and Anne C. Smith [2]

1. Department of Anesthesia, Massachusetts General Hospital, Boston, MA 02114, USA
2. Independent Researcher, Boston, MA 02114, USA
* Correspondence: daugust@mgh.harvard.edu; Tel.: +1-617-724-2250

**Abstract:** Cryptosystems employing a synchronous binary-additive stream cipher are susceptible to a generic attack called 'bit-flipping', in which the ciphertext is modified to decrypt into a fraudulent message. While authenticated encryption and message authentication codes can effectively negate this attack, encryption modes can also provide partial protection against bit-flipping. PudgyTurtle is a stream-cipher mode which uses keystream to encode (via an error-correcting code) and to encipher (via modulo-2 addition). Here, we describe the behavior of this mode during bit-flipping attacks and demonstrate how it creates uncertainty about the number, positions, and identities of decrypted bits that will be affected.

**Keywords:** error-correcting code; non-systematic code; symmetric encryption; stream cipher; encryption modes; malleability; integrity





## 1. Introduction

Among stream-cipher systems, the 'synchronized binary-additive stream cipher' (S-BASC) is the canonical example. In the S-BASC, a sequence of pseudo-random bits ('keystream') is generated by an algorithm acting on a finite-state machine ('keystream generator' or KSG) and then combined with the plaintext via bit-wise modulo-2 addition. The starting state of this KSG includes a secret key and often a randomizing initial value (IV; such as a nonce or frame number). To ensure full mixing of this initial state, the system may also require some number of KSG iterations before encryption begins ('warm-up').

Besides the synchronous (S-BASC) mode, stream-cipher systems can also utilize other modes, including a recently-described one called PudgyTurtle [1]. This mode operates on small groups of bits ('symbols') instead of individual bits and has the unusual property that the encryption requires an uncertain amount of keystreams: each plaintext symbol is *encoded* using a variable-length section of keystream, and then each codeword is *enciphered* by XOR'ing it to a separate, fixed-length section of keystream.

Some cryptanalytic techniques against stream ciphers target specific features of a particular KSG (e.g., distinguishing and correlation attacks) [2–4], while others are generic methods that work against any system (e.g., brute-force and time-memory tradeoff attacks) [5–9]. Here, we discuss another generic approach called the 'bit-flipping attack' (BFA), in which the opponent perturbs the ciphertext so that it will be decrypted into something other than intended. The BFA takes advantage of *malleability*—a major weakness of S-BASC systems which allows for the alteration of the ciphertext in ways undetectable by the receiver.

The BFA has two variations (*nonselective* and *tailored*), whose essential difference is the requirement for known plaintext. During a nonselective BFA, the attacker adds random bits to the ciphertext, thus making the decryption unrecognizable and rendering the channel useless. This variation is mentioned here only for completeness and is more suitably discussed in the context of jamming in communications theory or denial-of-service attacks in network security theory [10,11]. During a tailored BFA, known plaintext $X'$ is used to craft one or more bit-flips that will produce some desired decryption $X^*$. Let $\oplus$

represent modulo-2 addition (XOR), and let $Y'$ and $K'$ represent sections of ciphertext and keystream corresponding to $X'$. The opponent intercepts $Y' = X' \oplus K'$ and then transmits the modified ciphertext $Y^* = Y' \oplus X' \oplus X^*$. Since XOR is an involution, decryption of this 'flipped' ciphertext will produce

$$\begin{aligned} Y^* \oplus K' &= (Y' \oplus X' \oplus X^*) \oplus K' \\ &= (\,[X' \oplus K'] \oplus X' \oplus X^*) \oplus K' \\ &= X^* \end{aligned}$$

Standard defenses against bit-flipping include authenticated encryption and the message-authentication code (MAC)—a keyed hash of the plaintext which allows the recipient to detect tampering and reject nonauthentic ciphertexts (Chapter 4 [12]). Another strategy against bit-flipping is to use an encryption mode. In 2001, for example, Golić noted that stream-cipher modes with an 'inifite memory' have *"an inherent potential that can be used for message integrity purposes"* [13]. Modes that incorporate ciphertext feedback, for instance, could propagate the effects of changing even one ciphertext-bit, thereby making it difficult to tailor a BFA.

Although the connection between encryption modes and integrity pointed out by Golić has been reported in the literature for decades, little has been written about the actual details of using stream-cipher modes for this purpose and nothing at all about PudgyTurtle in this context. The primary goal of this paper is to explore the behavior of this new encryption mode against a BFA. We demonstrate that PudgyTurtle mode creates uncertainty about where exactly the decrypted text will start showing the effects of bit-flipping, how long these effects will persist, what the distribution of decrypted symbols will be, and whether or not length will be preserved. We emphasize that despite these interesting (and sometimes unique) features, PudgyTurtle provides only *partial* protection against bit-flipping. It reduces, but does not completely eliminate, ciphertext malleability.

The other contribution of this manuscript is to provide a flexible implementation of PudgyTurtle mode. To date, analysis of PudgyTurtle has focused on one particular version with nibble-sized (4-bit) input and byte-sized (8-bit) output. To study bit-flipping more broadly, however, we introduce a 'generalized' PudgyTurtle (denoted as PT[$s, f, d$]) which allows for variably sized symbols.

Our main results are as follows: (1) PudgyTurtle mode can be generalized, and its performance predicted; (2) the outcomes of all bit-flipping attacks against PudgyTurtle include an element of uncertainty; (3) some bit-flipping attacks against PudgyTurtle are rejected by the decryption algorithm, with a probability depending on the position of the flipped bit relative to its underlying codeword; (4) bit-flipping attacks can produce effects ranging from altering one symbol all the way to an 'avalanche'; (5) knowing in advance exactly how a BFA will alter the decrypted text is difficult, but generic statistical predictions are possible; and (6) some bit-flipping attacks may increase the length of decrypted text relative to the original plaintext.

After discussing notation (Section 2) and methods (Section 3), we review stream-cipher encryption modes (Section 4). Next, we present and analyze the generalized formulation of PudgyTurtle (Section 5) and use this more-flexible implementation to explore PudgyTurtle mode's behavior during bit-flipping attacks, including a (hypothetical) electronic banking fraud scheme (Sections 6 and 7).

## 2. Notation

### 2.1. Numbers

Hexadecimal values are prefixed by 0x, and binary values are subscripted by 2 (e.g., 254 can be written as 0xFE or $11111110_2$).

## 2.2. Functions and Operators

Operators include $\oplus$ for XOR (modulo-2 addition), $\otimes$ for AND (bitwise multiplication), $\gg$ for right-shift; $\|$ for concatenation, $\lfloor u \rfloor$ ($\lceil u \rceil$) for the floor (ceiling) of real-valued $u$, and $h(V)$ for the Hamming weight (i.e., number of 1 bits) in binary vector $V$. The Hamming distance ($\ell1$-norm) between binary vectors $V$ and $V'$ is $h(V \oplus V')$.

## 2.3. Symbols and Sequences

PudgyTurtle mode operates on small groups of bits ('symbols', denoted by uppercase letters) rather than individual bits (lowercase letters). Let $b_j$ represent $j$-th bit of binary sequence $B$, where $j = 1, 2, 3, \ldots, |B|$. The $i$-th (nonoverlapping) $s$-bit symbol within this sequence is

$$B[s]_i = b_{s(i-1)+1} \, \| \, b_{s(i-1)+2} \, \| \, b_{s(i-1)+3} \, \| \, \ldots \, \| \, b_{si}$$

where $i = 1, 2, 3, \ldots, N[s]_B$, and $N[s]_B = \lfloor |B|/s \rfloor$. For convenience, $[s]$ can be dropped in favor of the simpler notations $B_i$ and $N_B$ whenever the number of bits per symbol is unambiguous or irrelevant to the context.

$X$, $Y$, and $K$ stand for plaintext, ciphertext, and keystream, respectively. Symbols with primes are 'known' (e.g., the known plaintext and its corresponding ciphertext are represented by $X'$ and $Y'$). Symbols with asterisks occur during bit-flipping attacks (e.g., $Y^*$ is the modified ciphertext that has been subjected to bit-flipping, and $X^*$ is the decryption of this 'flipped' ciphertext).

## 2.4. Keystream Generator

PudgyTurtle is cipher agnostic: it takes bits from a 'black-box' KSG operating on an $n$-bit state $S$. The $t$-th keystream bit is $k_t = o(\pi(S_{t-1}))$, where $o : \{0,1\}^n \to \{0,1\}$ is the output function, $\pi : \{0,1\}^n \to \{0,1\}^n$ the state-update function, $t = 1, 2, 3, \ldots, |K|$, and initial state $S_0$ contains the secret key and IV.

## 3. Methods

In the experiments described below, the plaintext source (unless otherwise specified) is an 800,000-bit ASCII-encoded English-language text. Encryption is done via one of two 'toy' ciphers: (1) RC4 with a 40-bit key and (2) a simple, maximal-period 24-bit NLFSR with primitive polynomial $1 + x + x^8 + x^9 + x^{15} + (x^7 \cdot x^{18})$ [14]. We emphasize that these KSGs are chosen for simplicity and used for illustrative purposes only. Neither is intended as a practically secure stream-cipher, either with or without PudgyTurtle.

We study the simplest attack, in which a single ciphertext bit is flipped ($Y \to Y^*$), this modified ciphertext is decrypted ($Y^* \to X^*$), and this decryption is then compared to the original plaintext ($X^*$ vs. $X$). Measured outcomes include the following:

- $P_{REJECT}$ is the fraction of attacks for which the ciphertext is rejected. Rejection (described in detail later) occurs if bit-flipping produces an invalid codeword during decryption;
- $p(X^*)$ is the frequency distribution of values taken by all the decrypted symbols;
- $p(X_i^*)$ is the frequency distribution of values taken by the $i$-th decrypted symbol;
- $\mathrm{Hd}(i) = h(x_i \oplus x_i^*)$ is the Hamming distance between the $i$-th original plaintext bit and $i$-th bit in the decryption of $Y^*$;
- $\overline{\mathrm{Hd}}$ is the normalized Hamming distance between the original plaintext and decrypted (flipped) ciphertext:

$$\overline{\mathrm{Hd}} = \frac{\sum_i \mathrm{Hd}(i)}{|X|} = \frac{h(X \oplus X^*)}{|X|}$$

## 4. Stream-Cipher Modes

Here, we briefly review PudgyTurtle mode and also discuss stream-cipher modes in general.

### 4.1. PudgyTurtle Mode

In one implementation of PudgyTurtle, the plaintext is broken into 4-bit groups ('nibbles'), each of which is encrypted in four steps. First, an 8-bit *mask* is created by concatenating two nibbles of keystream. Next, keystream nibbles are generated until one of them *matches* the plaintext nibble—either exactly or to within a one-bit tolerance. This step produces two important quantities: the 'failure counter' (the number of keystream nibbles that failed to match the plaintext nibble) and the 'discrepancy code' (a number describing the relationship between the two matching symbols: 0 for an exact match or 1–4 to indicate the mismatched bit's position). This match is then *encoded* by concatenating the failure counter (taken as a 5-bit number) and the discrepancy code (taken as a 3-bit number), and the resulting codeword is *enciphered* by XOR'ing it with the mask.

There is one important question: What happens if so many failures ($>32$) occur during plaintext-to-keystream matching that their number can no longer be described with five bits? Whenever such an *overflow event* occurs, a special codeword is enciphered by XOR'ing it with the current mask and then inserting it into the ciphertext. After this, a new mask is created; the failure counter is reset to zero, and attempts to match the current plaintext nibble continue.

This special 'all-1' codeword, `0xFF`, comes from concatenating the maximum failure counter ($2^5 - 1 = 31 = 11111_2$) with one particular discrepancy code ($111_2$) reserved for overflows. Note that the maximum failure counter alone does not always indicate an overflow (e.g., an exact match between a plaintext nibble and the 32 nd keystream nibble would produce codeword $11111_2 \parallel 000_2 = $ `0xF8`, not the overflow codeword).

Decryption is a straightforward reversal of this procedure. First, a mask is built by concatenating two keystream nibbles. Next, the mask is XOR'ed with the current 8-bit ciphertext byte to *unmask* (decipher) the underlying codeword. This codeword is split into its failure counter (upper 5 bits) and discrepancy code (lower 3 bits). A number of keystream nibbles are generated (specifically, one more than the failure counter), the last one of which matches the plaintext nibble to within a bit. The discrepancy code is then used to 'reverse engineer' (decode) this keystream nibble back into the correct plaintext nibble; the result is output, and decryption of the next ciphertext byte commences.

If an overflow is detected during decryption (i.e., when unmasking produces the special all-1 codeword `0xFF`), then no decrypted symbol is output. Instead, 32 keystream nibbles are generated and discarded, a new mask is built, the *next* ciphertext byte is unmasked, and decryption continues as decsribed above.

### 4.2. Classical Modes: Synchronous and Asynchronous

A 'general binary stream cipher' operating without delay has state-update and encryption equations

$$S_{t+1} = \pi_{\text{KEY}}(S_t, x_t), \text{ and } y_t = x_t \oplus o_{\text{KEY}}(S_t)$$

where KEY is the initial KSG-state ($S_0$). Stream ciphers are classically described as having two encryption modes: *synchronous* (S-BASC) and *asynchronous* [15,16]. In the synchronous mode, the state-update function operates independently of plaintext or ciphertext: $S_{t+1} = \pi_{\text{KEY}}(S_t)$. In the asynchronous mode, this no longer holds. Golić further subclassified asynchronous systems into those with finite and infinite memory [13]. For the finite-memory type, designated as 'self-synchronizing stream ciphers', the KSG state itself includes feedback from $n$ previous ciphertext bits: $S_t = (y_{t-1}, y_{t-2} \text{ and } \ldots, y_{t-n})$. For the infinite-memory type, designated as 'stream cipher with memory', $\pi$ operates not only on $S_t$ but also on $x_t$.

More recently, Hamann, Krause, and Meier proposed *FP*(1)-mode, which has been instantiated in the lightweight stream cipher LIZARD [17,18]. *FP*(1)-mode—designed for packet-based operations in which each key/IV only generates a limited amount of keystream—consists of three phases: key loading (creating a KSG state from the secret key

and IV), key-mixing (repeatedly iterating this KSG-state while feeding back its nominal output), and key hardening (XOR'ing this mixed KSG-state with the secret-key).

Synchronous mode, asynchronous mode, and PudgyTurtle mode are illustrated in Figure 1. Here, 'key' is the initial KSG state (secret-key ± IV), black boxes represent the KSG, and the gray box stands for PudgyTurtle's error-correcting code (ECC). Figure 1A illustrates synchronous (S-BASC) mode, also called 'memoryless' and 'key-autokey'. This mode is used in numerous stream-cipher systems and by block ciphers operating in CTR (counter) mode.

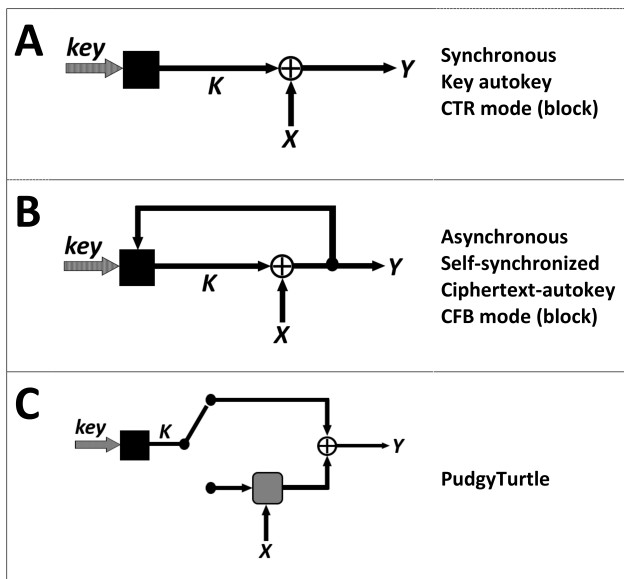

**Figure 1.** *Stream cipher modes.* Diagrams (**left**) and terminology (**right**) for various stream-cipher encryption modes. $X$ and $Y$ represent plaintext and ciphertext; the black boxes represent a keystream generator, which produces keystream $K$ from some initial state (key); and the gray box is an error-correcting code. (**A**) The synchronous mode follows the simple rule that $Y = X \oplus K$. (**B**) In the asynchronous mode, the keystream generator's state includes feedback from some previous ciphertext. (**C**) PudgyTurtle mode utilizes the keystream in two different ways. When the toggle is down, $K$ is used to encode a multibit plaintext 'symbol'. When the toggle is up (shown here), a different portion of $K$ is XOR'ed with this codeword to encipher it. The toggle remains in the down position for a variable duration so that the total amount of $K$ required to encrypt varies from symbol to symbol.

Figure 1B depicts the asynchronous mode (e.g., finite-memory type), also called 'ciphertext-autokey' and 'self-synchronizing'. Here, the KSG state incorporates some number of previous ciphertext bit(s). Errors from a flipped ciphertext bit 'wash out' after $n$ KSG updates (where $n$ is the KSG-state size), thereby resynchronizing the system. Asynchronous modes are used by the stream-ciphers SAVILLE (an older Suite-A system jointly designed by NSA and GCHQ) [19], PKZIP [20,21], Hiji-bij-bij [22], WAKE [23], the T-function system of Klimov and Shamir [24], and by block ciphers in the CFB (cipher feedback) mode.

Figure 1C shows PudgyTurtle mode. As suggested by the toggle-switch selector, this mode uses different segments of keystream for different purposes. Some keystream is used to encode a multi-bit plaintext symbol (when the toggle is 'down'), and another section of keystream is XOR'd to this codeword to encipher it (when the toggle is 'up'). PudgyTurtle shares some features with the synchronous mode (keystream can be generated in advance) and others with asynchronous mode (memory), but it also has unique features (the amount of keystream required to encrypt each plaintext symbol varies, $X$ and $K$ are not combined by a simple XOR, and 'memory' is not due to feedback from $Y$ to the KSG state but rather due to an iterative encoding process between $X$ and $K$).

### 4.3. Synchronization, Linkage, and Connection

Some new terminology is helpful to describe the relationships between plaintext, keystream, and ciphertext in PudgyTurtle mode. *Synchronized* is reserved for describing encryption modes themselves, not the sequences upon which they operate. *Linkage* is a low-level concept describing the mathematical relationship between plaintext, keystream, and ciphertext symbol indices. *Connection* is a high-level concept emphasizing that unambiguous decryption can be possible even for sequences that are not linked. During normal PudgyTurtle operation, $X$, $Y$, and $K$ are unlinked but connected. A BFA, however, may cause disconnection as well.

In more detail, *linked* describes two sequences whose identically indexed elements are functionally related. Figure 2a shows that during typical stream-cipher encryption, $X$, $K$, and $Y$ are linked: $y_i$ is a function of $x_i$ and $k_i$. Even with asynchronous modes, where $k_i$ may be a more complex function of previous ciphertext or plaintext bits, linkage is maintained. With PudgyTurtle mode, however, these sequences become unlinked (Figure 2b): $Y_j$ is now a function of $X_i$ and $k_t$—the first bit of the relevant section of keystream. Overflows cause the $j$ to drift ahead of $i$, and the uncertainty of each plaintext-to-keystream match causes $t$ to drift ahead of both $j$ and $i$. Despite being unlinked, however, the sequences remain connected, and so decryption is still possible.

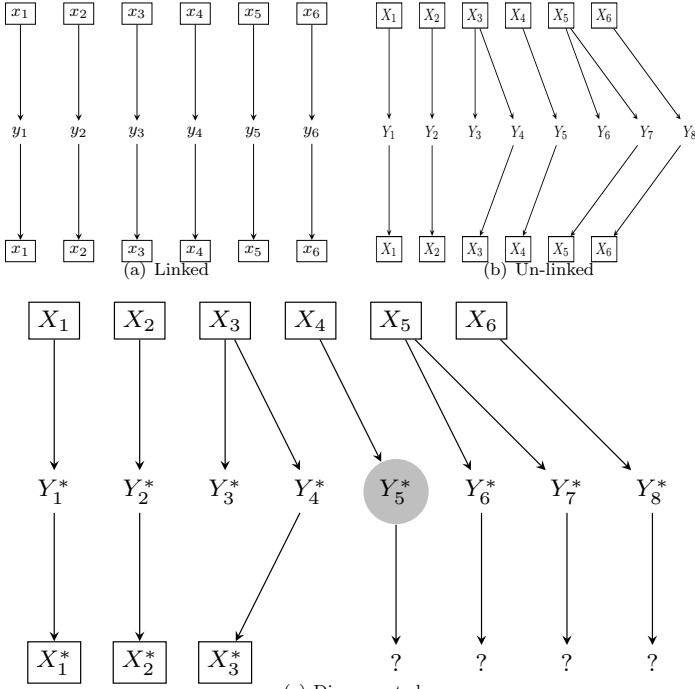

**Figure 2.** *Other encryption modes, PudgyTurtle, and bit-flipping.* (**a**) For non-PudgyTurtle encryption modes, each plaintext bit ($x_i$) is *linked* to its corresponding ciphertext bit ($y_i$) through the keystream bit $k_i$—whether the KSG state that generated $k_i$ includes feedback (asynchronous mode) or not (synchronous mode). (**b**) PudgyTurtle-mode *unlinks* the indices of plaintext symbol $X_i$, its corresponding ciphertext symbol $Y_j$, and the first bit of keystream used during its encryption, $k_t$. Overflows cause $j$ to drift ahead of $i$, and the inherent uncertainty in each plaintext-to-keystream match causes $t$ to drift ahead of $i$ and $j$. Despite being unlinked, unambiguous decryption of $Y_j$ into $X_i$ from keystream bit $k_t$ is still possible. (**c**) A bit-flipping attack against PudgyTurtle may *disconnect* the ciphertext ($Y^*$) from the plaintext. In this cartoon, bit(s) within the fifth ciphertext symbol (gray circle) have been flipped. If these flipped bits alter a failure counter, then the keystream symbol actually used during decryption (i.e., the one that supposedly 'matched' $X_4$) will be incorrect. Not only will disconnection affect how $Y_5^*$ gets decrypted but will also cause subsequent ciphertext symbols to be 'unmasked' into apparently random values ('?' symbols) rather than into the correct codewords.

During a bit-flipping attack, however, *Y* and *K* may *disconnect*. Figure 2c shows what happens when a flipped bit within the fifth ciphertext symbol (gray circle) causes $Y_5^*$ to be unmasked into a new, incorrect codeword whose failure counter differs from what is should be. Now, decryption no longer works: the PudgyTurtle algorithm 'thinks' that the wrong keystream symbol should be used to decrypt $Y_5^*$ into $X_4^*$. Disconnection affects not only this symbol but also triggers an avalanche effect: all subsequent ciphertext symbols will be unmasked into seemingly random values (?-symbols) rather than into correct codewords.

## 5. Generalized PudgyTurtle

To study bit-flipping attacks most broadly, it would be useful for PudgyTurtle to allow input and output symbols of many sizes—not just nibble-sized input and byte-sized output. Toward this end, we introduce the 'generalized' PudgyTurtle implementation PT[$s, f, d$]. Here, *s* is the size (in bits) of each plaintext symbol, *f* is the size of each failure counter, *d* is the size of each discrepancy code, and $c = f + d$ is the size of each codeword, mask, and ciphertext-symbol. In this notation, the original implementation was PT[4,5,3], which encrypted a 4-bit plaintext symbol into 8-bit ciphertext symbols.

### 5.1. Match Function

PudgyTurtle requires matching (to within a 1-bit tolerance) some *s*-bit plaintext symbol $X_i$ to some *s*-bit keystream symbol $K_F$. The details of this match are captured by a (0–*s*)-valued discrepancy code *D*, which can be expressed as the output of a *match function*, $\delta$: $\{0,1\}^s \rightarrow \{-1, 0, 1, 2, \ldots, s\}$:

$$\delta(X_i, K_F) = \begin{cases} 0, & \text{if } h(X_i \oplus K_F) = 0 \\ 1 + \log_2(X_i \oplus K_F), & \text{if } h(X_i \oplus K_F) = 1 \\ -1, & \text{if } 2 \leq h(X_i \oplus K_F) \leq s \end{cases}$$

For example, if the 5-bit plaintext and keystream symbols match everywhere except their fourth bit, then $h(X_i \oplus K_F) = 1$, $X_i \oplus K_F = 01000_2 = 8$, and so $D = \delta(X_i, K_F) = 1 + \log_2(8) = 4$. If, on the other hand, these two symbols match exactly, then *D* would be $h(0000_2) = 0$. Notice that $\delta() = -1$ is simply a 'place holder' for when $X_i$ and $K_F$ differ by >1 bit—this value will not become part of any discrepancy code.

### 5.2. Encryption

To encrypt the *s*-bit plaintext symbol $X_i$ into the *c*-bit ciphertext-symbol $Y_j$ with keystream starting at bit $k_t$, the following steps are taken:

1. **MASK**
   - Create a mask of *c* keystream bits: $M = (k_t \| k_{t+1} \| \ldots \| k_{t+c-1})$;
   - Update *t*, the new 'current' keystream bit: $t \leftarrow t + c$.

2. **MATCH** Starting from $k_t$, generate successive *s*-bit keystream symbols $K_0, K_1, K_2, \ldots$, until either

   (a) One of these keystream symbols, designated as $K_F$, matches $X_i$ exactly or differs from it by a single bit. In either case, proceed to Step 3;

   or ...

   (b) $2^f$ keystream symbols have failed to match. If this **overflow event** happens
      - Output the *c*-bit ciphertext-symbol $Y_j = (2^c - 1) \oplus M$;
      - Update the index of $k_t$: $t \leftarrow t + (2^f)s$;
      - Update the ciphertext-symbol index: $j \leftarrow j + 1$;
      - Return to Step 1.

3. **ENCODE** Make a *c*-bit *codeword*, $C = (F\|D)$, where failure-counter $F$ (the number of *s*-bit keystream symbols just tested against $X_i$ that failed to match) is represented by $f$ bits, and discrepancy code $D = \delta(X_i, K_F)$ is represented by $d$ bits.

4. **ENCIPHER**
   - XOR this codeword with its mask: $Y_j = C \oplus M$;
   - Output $Y_j$.

5. **UPDATE**
   - Update the keystream-bit index: $t \leftarrow t + (F+1)s$;
   - Update the plaintext-symbol index: $i \leftarrow i + 1$;
   - Update the ciphertext-symbol index: $j \leftarrow j + 1$;
   - Return to Step 1.

For example, consider encoding a plaintext-to-keystream match using PT[4,5,3], which has 4-bit symbols ($s = 4$), 5-bit failure counters ($f = 5$), and 3-bit discrepancy codes ($d = 3$), and produces 8-bit codewords ($c = 5+3 = 8$). Suppose plaintext-symbol $X_i$ matches the tenth keystream symbol against which it is tested everywhere except its high-order bit. Then $F + 1 = 10$, and so $K_F = K_9 = (k_{t+36} \| k_{t+37} \| k_{t+38} \| k_{t+39})$, and $D$ would be $\delta(X_i, K_9) = 1 + \log_2(1000_2) = 4$. The resulting codeword would be $C = (9\|4) = 01001_2 \| 100_2 = 01001100_2 = $ 0x4C, and thus the final ciphertext symbol would be 0x4C XOR'd with its corresponding 8-bit mask.

*5.3. Decryption*

To decrypt the *c*-bit ciphertext symbol $Y_j$ into $X_i$ using keystream starting at bit $k_t$, the following occurs:

1. **MASK**
   - Create a *c*-bit mask $M = (k_t\|k_{t+1}\| \ldots \|k_{t+c-1})$;
   - Update the initial keystream bit-index: $t \leftarrow t + c$.

2. **DECIPHER** *Unmask* the ciphertext symbol to reveal its underlying codeword $C = Y_j \oplus M$.

3. **OVERFLOW?** If $C$ is the 'all-1' overflow codeword, then
   - Generate and discard $2^f$ (*s*-bit sized) keystream symbols;
   - Update the keystream bit-index: $t \leftarrow t + (2^f)s$;
   - Update the ciphertext symbol index: $j \leftarrow j + 1$;
   - Return to Step 1.

4. **UNPACK** If $C$ was *not* the overflow codeword, then split it into two components: extract failure-counter $F$ from its $f$ highest-order bits and discrepancy-code $D$ from its $d$ lowest-order bits: $F = (C \gg d) \otimes (2^f - 1)$, and $D = C \otimes (2^d - 1)$.

5. **VALIDATE** If $D > s$, then halt the decryption and return $\bot$.

6. **DECODE** Use $F$ and $D$ to 'reverse engineer' the original plaintext-to-keystream match:
   - Generate $(F + 1)$ *s*-bit keystream symbols $K_0, K_1, K_2, \ldots, K_F$;
   - Recover the plaintext symbol from $K_F$ by inverting the discrepancy-code:

$$X_i = \begin{cases} K_F & \text{if } D = 0; \\ K_F \oplus 2^{D-1} & \text{if } 1 \leq D \leq s \end{cases}$$

7. **UPDATE**
   - Output $X_i$;
   - Update the index of $k_t$: $t \leftarrow t + (F+1)s$;
   - Update the ciphertext-symbol index: $j \leftarrow j + 1$;
   - Update the decrypted-symbol index: $i \leftarrow i + 1$;
   - Return to Step 1.

The VALIDATE step may seem redundant: at this point in the algorithm, discrepancy codes should always be in range (i.e., since the codeword is not an overflow, $D$ should

be between 0 and $s$). However, this step is made explicit because ciphertexts that have been subjected to bit-flipping attacks may produce *invalid* discrepancy codes at this point. In such cases, the ciphertext is rejected and no decryption is returned.

As an example (again using PT[4,5,3]), suppose that ciphertext $Y$ = 0xAB is deciphered by mask $M$ = 0xE7, producing the codeword $C$ = (0xAB $\oplus$ 0xE7) = 0x4C. Since this is not the overflow codeword (i.e., $C \neq$ 0xFF), we proceed with the unpacking step: 0x4C = $01001100_2$ = $01001_2 \parallel 100_2$ = $(9 \parallel 4)$ = $(F \parallel D)$. The validation step succeeds because the discrepancy code is within range (i.e., $D$ is not $> 4$). Next, since $F = 9$, we must generate ten new (4-bit) keystream symbols to reach the one that matched the current plaintext symbol. Since $D$ = 4, the plaintext symbol must have differed from $K_9$ in its fourth (high-order) bit: that is, $2^{D-1} = 2^3 = 8 = 1000_2$. Thus, the decrypted symbol is $X_i = K_9 \oplus 1000_2 = (1 \oplus k_{t+36}) \parallel k_{t+37} \parallel k_{t+38} \parallel k_{t+39}$.

*5.4. Indexing*

For a stream-cipher in synchronous mode, indexing is trivial: $y_i = x_i \oplus k_i$ for all $i$. For PudgyTurtle mode, this is not the case: it can be challenging to cross-index sequences or even to unambiguously index one sequence that may be split into different-sized symbols (e.g., $s$-bit and $c$-bit symbols within $K$).

Regarding $X$ and $Y$, although the symbol size for each sequence differs (i.e., $s$-bit groups for the plaintext, and $c$-bit groups for the ciphertext), this size remains fixed throughout encryption. Thus, both sequences can be indexed continuously:

$$X = X_1, X_2, X_3, \ldots, X_j, \ldots, X_{N[s]_X}$$

$$Y = Y_1, Y_2, Y_3, \ldots, Y_j, \ldots, Y_{N[c]_Y}$$

Because of overflows, $N[c]_Y$ may exceed $N[s]_X$, causing these two sequences to become unlinked: $Y_j$ no longer represents an encrypted version of $X_j$ but rather of $X_i$ where $i < j$. This issue is easily understood and just requires careful description of exactly what $'i'$ in $X_i$ (or $'j'$ in $Y_j$) means in a specific context.

Keystream indexing is more difficult because PudgyTurtle uses *different*-sized symbols within $K$ for different tasks (i.e., $c$-bit symbols for masks, and $s$-bit symbols to match the plaintext). For PT[4,5,3], it so happens that the each 8-bit mask is exactly twice the length of each 4-bit keystream (or plaintext) symbol. Thus, making a mask means 'concatenating two keystream symbols', a coincidence which allows continuous indexing of the entire keystream as 4-bit symbols.

For PT[$s, f, d$] however, the mask is not necessarily a concatenation of some whole number of $s$-bit symbols, nor is the index of the keystream bit at which matching begins necessarily a multiple of $s$. Thus, instead of indexing the entire keystream, only shorter keystream subsequences $K_0, K_1, K_2, \ldots, K_u, \ldots, K_F$ are indexed, where $0 \leq F < 2^f$ and simplified notation $K_u$ is used instead of $K[s]_u$. Each subsequence falls in between two masks. Assuming that the relevant section of keystream starts at $k_t$, then $K_u$ can be expressed in a precise but awkward way as follows:

$$K_u = k_{t+c+us} \parallel k_{t+c+us+1} \parallel \cdots \parallel k_{t+c+us+(s-1)}$$

However, since $t$ depends on the outcomes of all previous plaintext-to-keystream matches, each bit's index above becomes a history-dependent function of the plaintext and the secret key.

*5.5. Bit Padding*

In PT[$s, f, d$], the plaintext and ciphertext symbol lengths may not split evenly into groups of 8 bits. Thus, bit padding (specifically, Method #2 of ISO/IEC 9191-1) is employed so that input and output can be stored and displayed as bytes. In this padding technique, a single bit is appended to the original data, followed by zero or more 0 bits to achieve a context-specific total length. Bit padding during encryption is conducted in two steps:

- First, the plaintext is bit-padded to make a whole number of *s*-bit symbols;
- Next, the ciphertext is bit-padded to make a whole number of 8-bit bytes.

During decryption, an initial layer of bit-padding is removed from the ciphertext to obtain a whole number of *c*-bit symbols. After decryption into *s*-bit plaintext symbols, another layer of bit-padding is removed to obtain a whole number of 8-bit bytes.

*5.6. PT[s, f, d] Performance*

PudgyTurtle produces a bandwidth expansion called the *ciphertext expansion factor* (CEF) and also uses more keystream bits than plaintext bits (*keystream expansion factor* or KEF). CEF consumes memory, and KEF expends time. Since previous work described these measures only for PT[4,5,3], here we examine CEF and KEF for the generalized implementation and compare predictions to observed data.

Overflows

For PT[4,5,3], overflows are infrequent (one per ∼80,000 bytes). For PT[*s, f, d*], overflows can be more common, even occurring several times in a row. To describe this clearly, some new terminology is useful:

- An *overflow* means that attempts to match a plaintext symbol to $2^f$ successive keystream symbols have all failed;
- An *overflow event* is the occurrence of one or more overflows during encryption of a single plaintext symbol;
- The *order* of an overflow event is the number of overflows it contains: 1-order refers to a single overflow, 2-order refers to a double overflow, and so on. The case of 'no overflows' can also be formally described as a 0-order event. Note that for PT[4,5,3], all overflow events observed thus far are 1-order, so the total number of overflows equals the number of overflow events. When higher-order events occur, however, the number of overflows exceeds the number of events.

During plaintext-to-keystream matching, the chance of an exact match or one-bit mismatch between any two *s*-bit symbols is

$$p = \left( \frac{s+1}{2^s} \right)$$

In addition, the probability of a successful match after *m* failures is $(1-p)^m p$. Let discrete random variable $O \in \{0, 1, 2, \ldots\}$ represent the *order* of an overflow event. The probability of no overflows (i.e., a successful plaintext-to-keystream match within $2^f$ attempts) is

$$\Pr\{O = 0\} = \sum_{m=0}^{2^f - 1} (1-p)^m p$$

The probability of any overflows (i.e., an overflow event of order $\geq 1$) is

$$P_O = 1 - \Pr\{O = 0\} = \sum_{j=1}^{\infty} \Pr\{O = j\} \tag{1}$$

where the probability of each *j*-order event above is

$$\Pr\{O = j\} = \sum_{m=j \cdot 2^f}^{(j+1) \cdot 2^f - 1} (1-p)^m p$$

Figure 3 shows the observed (●) and predicted (○) overflow-event probabilities plotted against *s/f*. This X-axis parameter was chosen because overflows occur more often when symbols become bigger and/or fewer failures are allowed (larger *s/f*). The observed probabilities are from 121 different PT[*s, f, d*] implementations, obtained by systematically varying

$s$ and $f$ between 4 and 14, and – since the discrepancy-code does not affect $P_O$ – setting $d$ to the smallest value provides a sufficient number of codewords (i.e., $2^d \geq s + 2$). For each implementation, the plaintext was encrypted using a randomly keyed 24-bit NLFSR. Observed data (●) came from dividing the actual number of overflow events by the number of plaintext symbols. Predicted probabilities (○) came from substituting each $s$ and $f$ into (1). Notice that predictions are very similar to the observed data and that overflows increase with $s/f$ as anticipated. When $s/f < 1.25$, $P_O$ reaches a floor value, and when $s/f > 2$, overflows become common enough to dominate the behavior of PudgyTurtle mode.

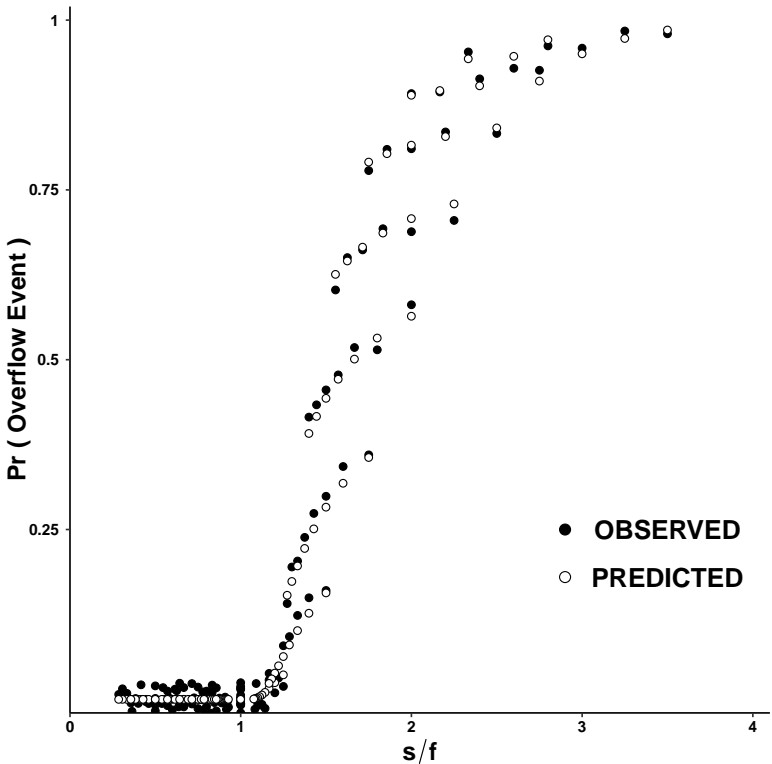

**Figure 3.** *Overflow events.* The probability of overflow events as a function of $s/f$, where $s$ is the number of bits per plaintext symbol, and $f$ is the number of bits per failure counter. Open symbols (○) are predicted using Equation (1). Filled symbols (●) represent observed data, tallying the number of overflow-events during randomly-keyed NLFSR encryption of an 800,000-bit message for many different PT$[s, f, d]$ implementations. Note that predictions and observations are similar and that overflow events are rare for small $s/f$, then become more common when $s/f > 1.25$ and reach a high-probability plateau when $s/f > 2$.

### 5.7. Expansion Factors

Since overflows were uncommon for PT[4,5,3], their effects on CEF and KEF can be ignored. For PT$[s, f, d]$, however, new expressions for these expansion factors are required to properly account for overflows.

*Observations.* Data were obtained from 1859 different PT$[s, f, d]$ implementations, in which $s$, $f$, and $d$ all ranged between 4 and 16. For each implementation, the same plaintext was encrypted using an NLFSR with a unique, randomly chosen key. CEF and KEF were then calculated as $|Y|/|X|$ and $|K|/|X|$ respectively, and plotted in Figure 4.

Figure 4A plots CEF against $s/f$—again using this X-axis parameter since 'more overflows' obviously also implies 'more ciphertext'. Interestingly, the CEF curve is U-shaped, with a minimum when $s/f \approx 1$. Above this, CEF increases because of more overflows (i.e., larger $s$ and/or smaller $f$ reduce the likelihood of each plaintext-to-keystream match). Below this, CEF increases not because of overflows but because of arithmetic: traveling leftward along the X-axis makes $s$ smaller, $f$ larger, or both, which in turn makes $c/s = (f + d)/s$—a

major determinant of CEF—bigger (see 'Predictions' below). This local minimum can be better appreciated in Figure 4B, which is a close-up of $0 < s/f < 2$. This graph also includes lines fitted to discrepancy-code sizes $d = 4$ and 12. These lines suggest that for a given $s/f$ ratio, increasing $d$ increases CEF—again illustrating its dependence on $(f + d)/s$.

KEF depends strongly upon $s$, as shown in Figure 4C. Once $s$ exceeds ~10, keystream expansion is in the hundreds, and encryption takes noticeably longer. Figure 4D provides a close-up view of KEF values for $4 < s < 7$, along with lines fitted to codeword lengths $c = 10$ and 15. These lines suggest that changing $c$ (which could be done by altering $f$, $d$, or both) affects KEF less than does changing $s$.

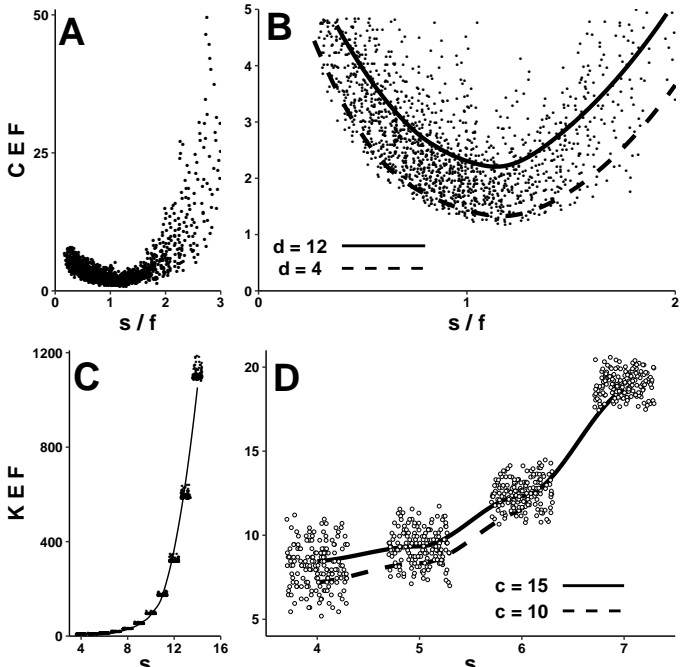

**Figure 4.** *Observed expansion factors.* CEF and KEF for >1800 NLFSR encryptions of the same plaintext, each using a unique secret key and a different PudgyTurtle implementation. Panel (**A**) depicts CEF as a function of $s/f$. Panel (**B**) shows a close-up of this data for $0 < s/f < 2$, along with fitted curves for $d = 4$ and 12. Notice that CEF is minimized when $s/f \approx 1$ and increases with $d$ for a given $s/f$. Panel (**C**) illustrates KEF as a function of $s$. Panel (**D**) provides a close-up view of KEF when $4 < s < 7$. The fitted lines here for codeword-sizes $c = 10$ and 15 (where $c = f + d$) suggest that PudgyTurtle's other two parameters have relatively less impact on KEF than does $s$. Abbreviations: CEF—ciphertext expansion factor; KEF—keystream expansion factor; NLFSR—nonlinear feedback shift register; $s$—plaintext-symbol size; $f$—failure-counter size; $d$—discrepancy-code size.

### 5.7.1. Predicted CEF

Regarding bandwidth (ciphertext) expansion, recall that CEF was 2 for PT[4,5,3] when overflows were ignored. This value was obtained by dividing the size of each ciphertext symbol ($c = 5 + 3 = 8$ bits) by the length of each plaintext symbol ($s = 4$ bits). Thus, without overflows, it would be expected that CEF = $c/s$ for PT[$s$, $f$, $d$].

The total number of overflows is the number of overflow events ($N[s]_X \cdot P_O$) multiplied by the typical number of overflows per event. With this latter quantity, the expected value of random variable $O$, is

$$E_O = \sum_j j \cdot \Pr\{O = j\} = \Pr\{O = 1\} + 2\Pr\{O = 2\} + 3\Pr\{O = 3\} + \ldots \qquad (2)$$

Since each overflow adds one more $c$-bit symbol to the ciphertext, all the overflows together will add $c \times (N[s]_X \cdot P_O) \times E_O$ more bits in total. Thus, a more accurate prediction of ciphertext expansion would be

$$\text{CEF} = \frac{|Y|}{|X|} = \frac{1}{|X|}(c \cdot N[s]_X + c \cdot N[s]_X \cdot P_O E_O)$$
$$= \left(\frac{c}{s}\right) \times (1 + P_O E_O) \tag{3}$$

where $N[s]_X = |X|/s$.

With $E_O$ above as an infinite sum, how many of its terms should be used when calculating CEF? Recall that $P_O$, the probability of an overflow event in Equation (1) can be calculated two different ways. One way (the complement of the no-overflow probability) yields its definitive 'true' value. The other way approximates $P_O$ with an infinite sum. We first determine how many terms are required to make this summation converge to within $10^{-8}$ of the true $P_O$ and then use this same number of terms to calculate $E_O$ in Equation (2) as well.

5.7.2. Predicted KEF

Regarding keystream expansion, recall that KEF was 5.2 for PT[4,5,3] when overflows were ignored. This value was obtained by adding the number of keystream symbols per mask (2, the number of 4-bit nibbles in an 8-bit mask) to the average number of keystream symbols required for a successful plaintext-to-keystream match (16/5 = 3.2, the mean of a geometric distribution with $p = (s+1)/2^s = 5/16$). Thus, without overflows, it might be expected that KEF = $c/s + 1/p$ for PT[$s, f, d$].

To account for overflows, consider the keystream as being composed of two (noncontiguous) parts: $K_O$ includes all the bits consumed by overflow events and $K_{match}$ includes all the bits consumed by successful (nonoverflow) plaintext-to-keystream matches. Thus, KEF = $|K|/|X| = (|K_O| + |K_{match}|)/|X|$. To determine $|K_O|$, note that each overflow consumes one $c$-bit mask plus $2^f$ symbols or $c + s(2^f)$ keystream bits total. This number, multiplied by the total number of overflows, will equal $|K_O|$:

$$|K_O| = N[s]_X \cdot P_O \cdot E_O \cdot (c + (2^f)s)$$

To determine $|K_{match}|$, let $\langle\#\rangle$ represent the average number of keystream symbols required for a match. The amount of keystream needed to represent all the matches will be

$$|K_{match}| = N[s]_X \cdot (c + s\langle\#\rangle)$$

Combining these two quantities and using $\langle\#\rangle = 1/p = 2^s/(s+1)$ as mentioned earlier, we acquire the new expression for KEF as follows:

$$\text{KEF} = \frac{|K|}{|X|} = \frac{|K_{match}| + |K_O|}{|X|}$$
$$= \frac{1}{|X|}N[s]_X\left(c + s\langle\#\rangle + P_O E_O(c + (2^f)s)\right)$$
$$= \frac{c}{s} + \frac{1}{p} + P_O E_O(\frac{c}{s} + 2^f)$$
$$= \frac{c}{s}(1 + P_O E_O) + \frac{1}{p} + P_O E_O 2^f \tag{4}$$

Before examining the accuracy of this formula, however, one point must be addressed. In theory, the average number of keystream symbols needed for a successful plaintext-to-keystream match is $\langle\#\rangle = 1/p$. However, in practice, $\langle\#\rangle \leq 2^f$ since an overflow is triggered after $2^f$ failures. Thus, if $1/p \gg 2^f$, KEF predicted by Equation (4) will exceed the observed KEF.

Figure 5 illustrates the accuracy of CEF and KEF predictions according to the above data from 1859 different PT[$s, f, d$] implementations. Panel A plots the predicted (Y-axis) vs. the observed (X-axis) CEF for values $< 25$. As expected, points generally fall along the identity line. Panel B is a similar plot for KEF values $< 50$. Most points fall along the identity, but some predictions overestimate the observed data (arrows above the identity line). These prediction errors should be minimal when $1/p$ is small compared to $2^f$ and become increasingly obvious as $1/p$ gets larger and eventually surpasses $2^f$. Panel C shows the relative error (i.e., $\Delta(\text{KEF}) = (\text{KEF}_{predicted} - \text{KEF}_{observed})/\text{KEF}_{predicted}$ plotted against a dimensionless parameter

$$\log_2 \frac{(1/p)}{2^f} = \log_2 \frac{2^{s-f}}{s+1}$$

When $(1/p) = 2^f$, their ratio is 1, and this parameter is zero (vertical line at X = 0). Negative X values represent predictions when $1/p$ is small compared to $2^f$, and positive X values represent predictions in the converse situation. As expected, the error in predicting KEF is small when $1/p \ll 2^f$ and becomes substantial when $1/p \gg 2^f$. A transition zone of increasingly inaccurate predictions extends $\sim \pm 1$–2 log-units from the vertical line.

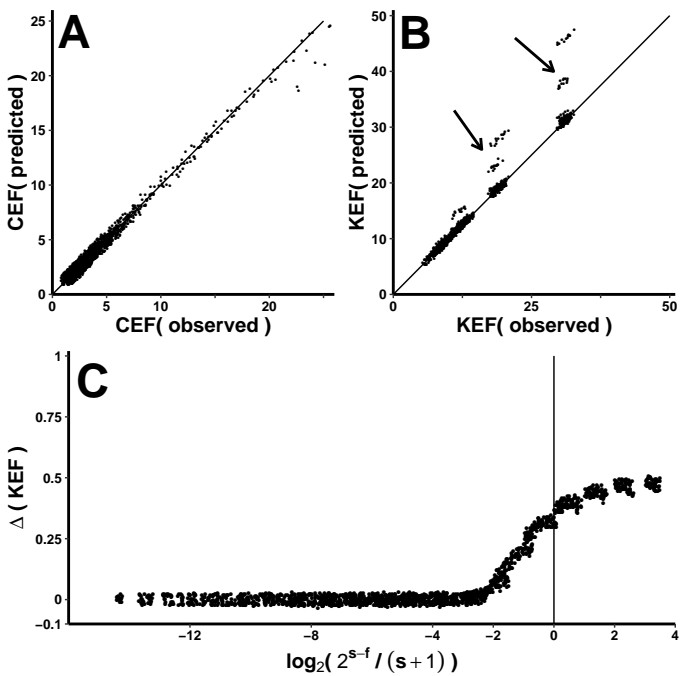

**Figure 5.** *Observed and predicted expansion factors.* Observed and predicted bandwidth expansion (CEF) and keystream expansion (KEF) were compared using encryption data from >1800 different PT[$s, f, d$] implementations. (**A**) Predicted CEF (Y-axis) vs. observed CEF (X-axis) for CEF values $< 25$. Most points fall along the identity line. (**B**) In this similar plot for KEF $< 50$, notice that predictions sometimes overestimate the observations (arrows). (**C**) The error in predicting KEF (i.e., $\Delta(\text{KEF}) = (\text{predicted} - \text{observed})/\text{predicted}$) is plotted against the base-2 logarithm of $(1/p)$ divided by $2^f$, where $p = (s+1)/2^s$ is the probability of two $s$-bit symbols matching to within a one-bit tolerance. The vertical line at X = 0 indicates a ratio of one, meaning that these two quantities are equal. The error is low when $(1/p)$ is small compared to $2^f$ (negative X-axis) and higher once $(1/p)$ exceeds $2^f$ (positive X-axis). Abbreviations: CEF—ciphertext-expansion factor; KEF—keystream-expansion factor; $s$—plaintext-symbol size; $f$—failure-counter size; $d$—discrepancy-code size.

## 6. PudgyTurtle and Bit-Flipping Attacks

We investigate the simplest possible BFA, in which one ciphertext bit is flipped, the modified ciphertext ($Y^*$) is decrypted, and this decryption ($X^*$) is compared against the original plaintext ($X$). Insight into how this mode behaves during such attacks is gained

by varying the PudgyTurtle implementation parameters $s$, $f$, and $d$, and by changing the position of the flipped bit (within $Y$ as a whole and within any given $c$-bit symbol).

### 6.1. Rejected Ciphertexts

One important concept is that a flipped ciphertext bit may produce an *invalid* discrepancy-code during decryption. When this occurs, the ciphertext will be rejected: the decryption algorithm halts at Step #5 (VALIDATE) and returns $\perp$. For instance, PT[4,5,3] has $2^3 = 8$ possible discrepancy codes, but only six are actually assigned:

| $000_2$ | $001_2$ | $010_2$ | $011_2$ | $100_2$ | $101_2$ | $110_2$ | $111_2$ |
|---------|---------|---------|---------|---------|---------|---------|---------|
| | Valid (to encode plaintext-to-keystream matches) | | | | **NOT valid** | **NOT valid** | Valid (to encode overflows) |

Flipping a bit could cause the new ciphertext symbol to be 'unmasked' into an invalid codeword containing $D = 101_2$ or $110_2$ (both invalid) or containing $D = 111_2$ (valid) paired with a failure counter $F \neq 11111_2$. In all, PT[4,5,3] has 95 invalid (8-bit) codewords, with 32 ending in $101_2$, 32 ending in $110_2$, and 31 more ending in $111_2$ but not beginning with $11111_2$—none of which would allow $K_F$ to be converted back into a plaintext-symbol.

### 6.2. Two Categories of Tailored BFA

Each ciphertext symbol is created by XOR'ing a codeword to a same-sized mask of keystream. In turn, each codeword is a failure counter concatenated with a discrepancy code ($F\|D$), including the special overflow case, where $F$ and $D$ are both 'all-1'. Symbolizing bit #$u$ of ciphertext-symbol #$j$ as $y_u^j$,

$$
\begin{aligned}
Y_j &= (\; y_{(j-1)c+1} \,\|\, y_{(j-1)c+2} \,\|\, \cdots \,\|\, y_{jc} \;) \\
&= (\; y_1^j \,\|\, y_2^j \,\|\, \cdots \,\|\, y_c^j \;) \\
&= \underbrace{(\; y_1^j \,\|\, y_2^j \,\|\, \cdots \,\|\, y_f^j \;)}_{\substack{\text{Enciphered} \\ \text{Failure counter}}} \,\|\, \underbrace{(\; y_{f+1}^j \,\|\, y_{f+2}^j \,\|\, \cdots \,\|\, y_{f+d}^j \;)}_{\substack{\text{Enciphered} \\ \text{Discrepancy code}}}
\end{aligned}
$$

Let $y_b^j$ denote the bit within $Y_j$ that is flipped during an attack. Depending upon its position, there are two categories of BFAs:

FLIP-F attacks alter a failure-counter: $y_1^j \leq y_b^j \leq y_f^j$;

FLIP-D attacks alter a discrepancy-code: $y_{f+1}^j \leq y_b^j \leq y_{f+d}^j$.

These two kinds of attacks have qualitatively different effects. During a FLIP-F attack, the keystream and ciphertext usually become disconnected, causing an *avalanche* effect: many bits of $X^*$ and $X$ will differ. In contrast, during a FLIP-D attack, the keystream and ciphertext usually remain connected: only a few bits of $X^*$ and $X$ will differ. Here, 'avalanche' means simply that flipping one ciphertext bit changes about half of the decrypted bits. We do not claim that a 'strict avalanche criterion' (SAC) is satisfied since this requires formal statistical testing of all input/output bit combinations. The reason for not using SAC is that avalanches often contain invalid discrepancy codes. In such cases, decryption returns $\perp$, and no data are available for statistical inference.

### 6.3. Localizing the Effect of a Flipped Bit

Encryption modes can affect the number and position of bits in $X^*$ that differ from their corresponding bits in $X$. For synchronous mode, flipping $y_b$ affects only $x_b^*$. For asynchronous/self-synchronizing mode, flipping $y_b$ affects the decryption starting $x_b^*$, and these

changes persist for $n$ bits (i.e., the KSG size). For PudgyTurtle mode, flipping $y_b$ can lead to variability in both the starting point and duration of these changes.

To begin studying these positional effects, a 256-bit message (extracted from the longer ASCII English plaintext) was encrypted under PT[8,4,4] using RC4 and a randomly generated key. One ciphertext bit (specifically one of the eight bits between $y_{128}$ and $y_{135}$) was then flipped. The result was decrypted, and the first and last bit index where $X^*$ and $X$ differed was tabulated. Enough secret keys were chosen to produce 100 'successful' attacks against each of the eight bit positions (i.e., 27,749 total attacks to obtain 800 that were not rejected). Figure 6A illustrates the *first* bit index at which $X^*$ and $X$ differed (Y-axis) for each flipped ciphertext bit on the X-axis. Figure 6B, organized similarly, shows the *last* bit index at which the decryption and original plaintext differed.

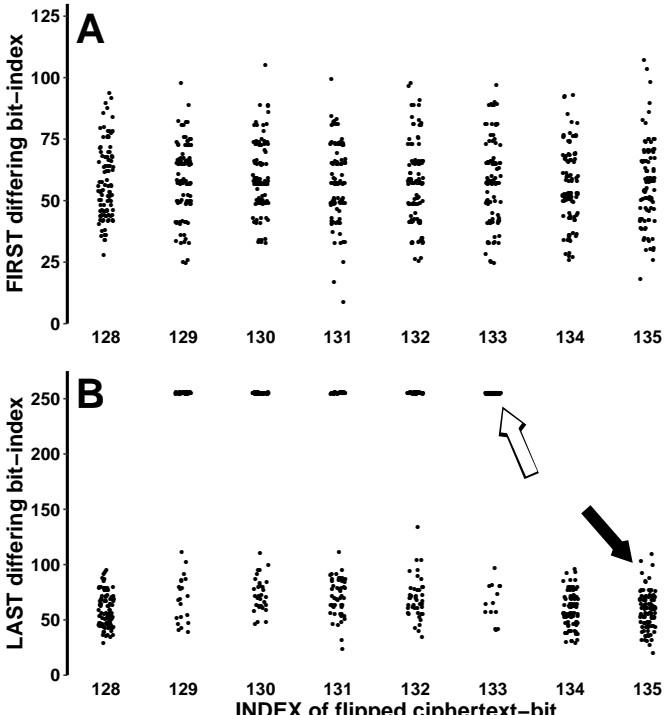

**Figure 6.** *Variable effects of flipping a single ciphertext bit.* Using PT[8,4,4], a 256-bit plaintext was RC4-encrypted under a randomly generated key. For each encryption, one ciphertext bit (between bit positions #128 and #135, shown on the X-axis) was flipped. A sufficient number of attacks were mounted to produce 100 decryptions at each bit position. Then, each decryption ($X^*$) and the original plaintext ($X$) were compared via a bit-wise Hamming distance measure $Hd(i) = h(x_i^* \oplus x_i)$. (**A**) The index of the first bit at which the decryption and plaintext differ (the smallest $i$ for which $Hd(i) = 1$). (**B**) The last bit index at which $X^*$ and $X$ differ (the largest $i$ for which $Hd(i) = 1$). For some attacks, the effect of bit-flipping continues throughout the entire message (white arrows near bit #256), while in other cases, the effect is more limited (solid arrow near bit #100). PudgyTurtle mode adds uncertainty about exactly where $X^*$ will start to diverge from $X$ and for how long this change will persist.

These results emphasize that PudgyTurtle mode makes it harder to predict the effect of flipping even a single ciphertext bit: the bit position at which changes in $X^*$ are first observed (Figure 6A) varies considerably as does the total number of affected bits (Figure 6B). The attacker can guess *on average* where and how many bits of $X^*$ might change but would find it hard to know the *exact* effects of flipping a particular ciphertext bit. PudgyTurtle also affects qualitative attack outcomes. For example, flipping some ciphertext bits (#129 through #133) can affect the whole decryption—as shown in Panel B by the points scattered near bit #256 on the Y-axis—while flipping other bits (#128, #134, or #135), only affects a limited segment of the decryption—typically not much beyond bit #100.

### 6.4. Positional Effects within a Codeword

The position of the flipped bit within a codeword (FLIP-F vs. FLIP-D) also affects the outcome of a BFA. Compared to flipping a discrepancy-code bit, flipping a failure-counter bit more often triggers an avalanche, increasing the chance of an invalid discrepancy code and leading to higher $P_{REJECT}$. In one example using PT[6,5,3] and RC4 key 0x1122334455, we observed the rejection rate for eighty BFAs (produced by flipping each bit from the first ten ciphertext symbols). Overall 63.8% of BFAs were rejected, but the flipped bit's position had a substantial effect: 96% of FLIP-F attacks were rejected, compared to only 10% of the FLIP-D attacks.

This suggests that for a given plaintext symbol size *s*, changing *f* and *d* while keeping codeword length *c* fixed should influence $P_{REJECT}$ predictably: larger *d* means more 'unassigned' discrepancy codes, creating more opportunities for invalid discrepancy codes to inadvertently appear during a BFA, ending with a higher rejection rate. This was confirmed using the same protocol as above, but using PT[6,5,*d*] with *d* = 3, 4, 5, and 6. Figure 7 shows that $P_{REJECT}$ does not change much among FLIP-F attacks (dotted line), which are rejected with high probability. Most of the variation in $P_{REJECT}$ occurs during FLIP-D attacks (dashed line), for which rejections rise from 10% with the lowest *d*-value to 56.7% with the highest *d*-value. Overall, the rejection rate was in the 65–75% range (solid line).

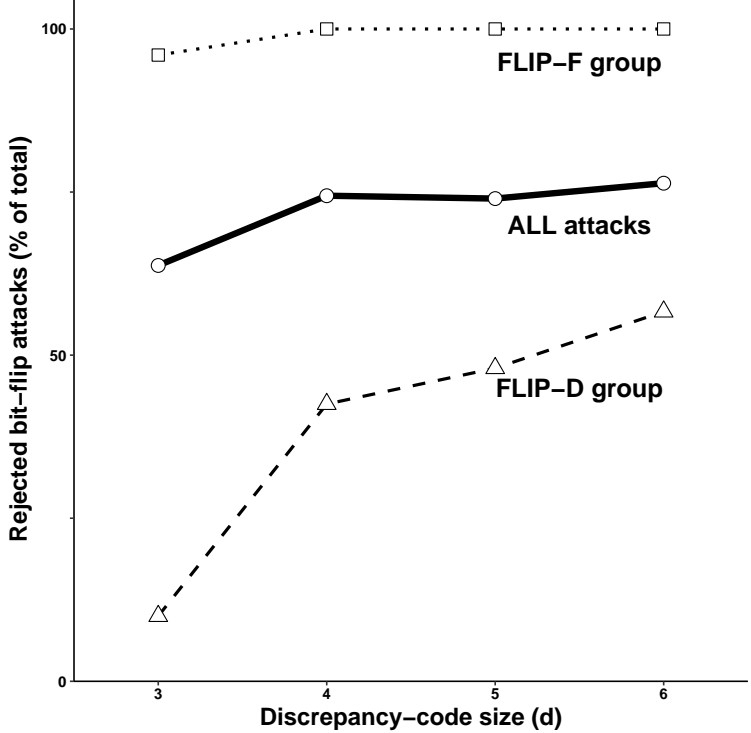

**Figure 7.** *BFA outcomes are position-dependent.* Bit-flipping attacks were mounted against PudgyTurtle implementations PT[6,5,*d*] with *d* = 3, 4, 5, and 6 by flipping each bit from the first ten ciphertext symbols. The Y-axis shows the fraction of attacks in which the ciphertext was rejected. Overall ∼60–75% of all attacks failed (solid line, 'ALL attacks'). Most attacks in which a failure counter was altered were rejected regardless of whether *d* was (FLIP-F, dotted line). Attacks in which a discrepancy code was altered were rejected less often and showed more dependence on *d* (FLIP-D, dashed line). Abbreviations: BFA—bit-flipping attack; FLIP-F—attacks in which a failure counter is altered; FLIP-D—attacks in which a discrepancy code is altered; *d*—discrepancy-code size.

Is this result somehow specific to PT[6,5,*d*] or does it generalize to other PudgyTurtle implementations? This was tested using the same RC4 key and again by flipping each bit within the first ten ciphertext-symbols. This time, however, all legal PT[*s*, *f*, *d*] implementations with *s*, *f*, and *d* ∈ {4, 5, 6, 7, 8} were used, producing a total of 15,000 separate

BFAs. Overall 79.32 ± 4.57% (mean ± s.d.) of ciphertexts were rejected. Among FLIP-F attacks, more were rejected more reliably (99.72 ± 1.33%) than among FLIP-D attacks (57.25 ± 13.33%). Together, these observations illustrate how the BFA outcome is affected by the relative position of the flipped bit within any *c*-bit ciphertext symbol and by the system parameters themselves.

*6.5. Predicting $P_{REJECT}$*

Since $P_{REJECT}$ is consistently high for FLIP-F attacks, a more interesting question is whether it can be predicted for FLIP-D attacks. With two examples, we demonstrate that specific attacks can be analyzed individually, but there does not appear to be an all-purpose predictive formula.

These examples use PT[8,5,4] and a 24-bit NLFSR, with 1000 unique, randomly chosen secret keys. For each encryption, attacks were performed against bit $y_6$, $y_7$, $y_8$, and $y_9$ (the ciphertext bits associated with the first discrepancy-code). Although all of these attacks are FLIP-D, their effects on $P_{REJECT}$ differ. Just over half (56%) of the attacks against bit $y_6$ were rejected, but only about one-third (36%) of attacks against the other three ciphertext bits were rejected.

Table 1 illustrates how these four attacks affect discrepancy codes. Columns 1 and 2 list the original discrepancy code (*D*) in decimal and binary. Of $2^d$ = 16 possible values, one ($D = 1111_2$) is for overflows, nine others ($D = 0$ through 8) describe plaintext-to-keystream matches, and the remaining six (gray background) are unassigned and would not occur during encryption. Columns #3–6 show the new discrepancy codes ($D^*$) after flipping ciphertext bits #6, 7, 8, and 9 respectively. Again, the gray background means that $D^*$ is invalid, and '-' stands for discrepancy codes that could not be produced during a BFA (i.e., because the original *D* from which $D^*$ is derived would never occur). One entry in this table deserves special mention. In Column 3 (Flip $y_6$), the parentheses around $D^* = 1111_2$ mean that this all-1 discrepancy code only produces a valid codeword when paired with the all-1 failure counter $F = 11111_2$. In this case, the resulting codeword is decodable as 'overflow'. If $D^* = 1111_2$ is paired with any other failure counter, however, the resulting codeword will be invalid, and the attack will be rejected.

To predict $P_{REJECT}$, our analysis assumes that $D = 1111_2 = 15$ is identical to an overflow event, and so $\Pr\{D = 15\} = P_O = 0.3181$ from (1). Additionally, we assume that all $(s + 1) = 9$ valid nonoverflow discrepancy codes are uniformly distributed with probability $p_D$—unlike failure counters, which are geometrically distributed. Since probabilities sum to unity, $p_D = (1 - P_O)/9 \approx 0.076$. Another useful probability (used shortly) is $\Pr\{F = 11111_2\}$, the failure counter that can occur either as part of an 'all-1' overflow codeword (when paired with $D = 1111_2$) or as part of a codeword representing a successful plaintext-to-keystream match to $K_{31}$ (when paired with $D \in \{0, 1, \ldots, 8\}$). Thus, $\Pr\{F = 11111_2\} = P_O + (1 - p)^{31}p = 0.3297$, where $p = (s + 1)/2^s = 9/256 = 0.0352$ is the usual probability of a successful plaintext-to-keystream match.

Consider first the FLIP-D attack against bit $y_6$, shown in Column 3 of Table 1. The original codeword in *Y* can contain any of ten discrepancy-codes ($D = 0$–8 and 15), while the new codeword in $Y^*$ can contain only four ($D^* = 0, 7, 8,$ and 15). Thus,

$$\Pr\{D^* = 0000_2\} = \Pr\{D = 1000_2\} = p_D$$
$$\Pr\{D^* = 1000_2\} = \Pr\{D = 0000_2\} = p_D$$
$$\Pr\{D^* = 0111_2\} = \Pr\{D = 1111_2\} = P_O$$
$$\Pr\{D^* = 1111_2\} = \Pr\{D = 0111_2 \cap F = 11111_2\} = (0.3297)p_D$$

Three of these new discrepancy codes ($D^* = 0, 7,$ or 8) would cause a codeword to be decoded as 'successful plaintext-to-keystream match'. The remaining one ($D^* = 15$) would cause a codeword to be decoded as 'overflow' (when paired with failure counter $F = 11111_2$) or as 'invalid' (when paired with any other failure counter).

**Table 1.** *FLIP-D attacks against the first ciphertext symbol.* For PT[8,5,4], this table shows possible outcomes of flipping each ciphertext bit corresponding to the first discrepancy code ($y_6$, $y_7$, $y_8$, and $y_9$). Columns 1 and 2 show the original discrepancy code ($D$) in decimal and binary; and Columns 3, 4, 5, and 6 show the new discrepancy codes ($D^*$) produced by flipping the first, second, third, and fourth bit of $D$, respectively. Gray indicates invalid codes, and '-' indicates codes that would not occur. One special case (parentheses: eighth row, third column) occurs when the first bit of $D = 0111_2$ is flipped, producing $D^* = 1111_2$. The new codeword ($F \parallel 1111_2$) is only valid if the failure counter happens to be $F = 11111_2$. In this case, the codeword represents an incorrect but valid 'overflow'; otherwise, the attack will be rejected. FLIP-D attacks behave differently depending upon which bit is flipped (e.g., flipping $y_6$ produces four possible $D^*$'s and allows for overflows, while attacks against $y_7$, $y_8$, and $y_9$ produce eight $D^*$'s but no overflows).

| Original | | Flip $y_6$ | Flip $y_7$ | Flip $y_8$ | Flip $y_9$ |
| $D$ | | $D^*$ | $D^*$ | $D^*$ | $D^*$ |
|---|---|---|---|---|---|
| 0 | 0000 | 1000 | 0100 | 0010 | 0001 |
| 1 | 0001 | 1001 | 0101 | 0011 | 0000 |
| 2 | 0010 | 1010 | 0110 | 0000 | 0011 |
| 3 | 0011 | 1011 | 0111 | 0001 | 0010 |
| 4 | 0100 | 1100 | 0000 | 0110 | 0101 |
| 5 | 0101 | 1101 | 0001 | 0111 | 0100 |
| 6 | 0110 | 1110 | 0010 | 0100 | 0111 |
| 7 | 0111 | (1111) | 0011 | 0101 | 0110 |
| 8 | 1000 | 0000 | 1100 | 1010 | 1001 |
| 9 | 1001 | - | - | - | - |
| 10 | 1010 | - | - | - | - |
| 11 | 1011 | - | - | - | - |
| 12 | 1100 | - | - | - | - |
| 13 | 1101 | - | - | - | - |
| 14 | 1110 | - | - | - | - |
| 15 | 1111 | 0111 | 1011 | 1101 | 1110 |

The probability that this BFA will *not* be rejected is the sum of the four values above, $P_{ACCEPT} = p_D(2 + 0.3297) + P_O = 0.4709$, which is similar to the observed value of 48.5%. The probability that the attack will be rejected is

$$P_{REJECT} = \Pr\{9 \leq D^* \leq 14\} + \Pr\{D^* = 1111_2 \cap F \neq 11111_2\}$$
$$= \Pr\{1 \leq D \leq 6\} + \Pr\{D = 0111_2 \cap F \neq 11111_2\}$$
$$= 6p_D + p_D \Pr\{0 \leq F < 32\})$$
$$= p_D(6 + \sum_{i=0}^{31}(1 - p)^i p)$$
$$= 0.5053$$

which is again similar to the observed value of 51.5%.

FLIP-D attacks against $y_7$, $y_8$, and $y_9$ all behave similarly to one another but differently from the attack against $y_6$. As a specific example, consider the attack against $y_9$. Applying the same reasoning to the last Column of Table 1,

$$P_{ACCEPT} = \Pr\{0 \leq D^* \leq 7\}$$
$$= \Pr\{0 \leq D \leq 7\}$$
$$= 8p_D = 0.6061$$

and

$$P_{REJECT} = \Pr\{D^* = 1001_2\} + \Pr\{D^* = 1110_2\}$$

$$= \Pr\{D = 1000_2\} + \Pr\{D = 1111_2\}$$
$$= p_D + P_O = 0.3939$$

Again, both predictions are similar to the observed values of 59.2% accepted and 40.8% rejected.

These two examples (BFAs against $y_6$ and $y_9$) show the mechanistic steps involved in calculating $P_{REJECT}$. While a similar approach can be applied to any FLIP-D attack against any PT[$s, f, d$], the details matter: there is no all-purpose formula, and the ultimate probability of interest depends on various inter-relationships between the position of the flipped bit within a $c$-bit codeword and the system parameters $s$, $f$, and $d$.

*6.6. Decrypted Symbol Frequencies*

How exactly will the decrypted bits change? In other words, how does the distribution of decrypted symbols $p(X^*)$ compare to that of the original plaintext $p(X)$? Since most FLIP-F attacks will be rejected, again, it is easier to explore this question using FLIP-D attacks. In this section, we present two experiments: one examines $p(X^*)$ the entire decryption, while the other is limited to one decrypted symbol $p(X_1^*)$.

*EXPERIMENT 1.* A 16-byte test pattern was encrypted with RC4 under 100 different secret keys. PudgyTurtle implementation, PT[8,4,4], was chosen to make each decrypted symbol one byte ($s = 8$ bits), so that its value can be plotted on a 16 × 16 grid with the Y (or X)-axis as its high (or low)-order nibble. The plaintext $X = $ 0x00112233 44556677 8899AABB CCDDEEFF was chosen to visually stand out on this grid as a diagonal line running from lower left (0,0) to upper right (0xF,0xF). For each ciphertext, every possible FLIP-D attack was carried out (i.e., flipping bits $y_5$–$y_8$ within $Y_1$, flipping bits $y_{13}$–$y_{16}$ within $Y_2$, and so on). For each attack, $Y^*$ was decrypted, and the identity of each decrypted byte tabulated.

Figure 8 shows a 'heat map' histogram of the observed distribution $p(X^*)$. Bytes matching the original plaintext are set off as white squares surrounded by dotted lines. On this diagonal (lower left to upper right), white is for visualization only—not for representation of any particular probability. For the off-diagonal squares, byte frequencies are represented via a normalized gray scale, where light gray stands for a frequency of ~0.05%, and the black for ~1.19%. (Note: a frequency of 0.4% would be expected for 250 uniformly distributed bytes.) Notice that the histogram does not appear to be completely uniform (e.g., the 4 × 4 squares in the upper left and lower right seem less common than do other similarly sized areas). What is more important than these details, however, is the fact that every off-diagonal cell is shaded to at least some degree: although the original plaintext distribution $p(X)$ contained only sixteen bytes, $p(X^*)$ contains all $2^8 = 256$ possible bytes.

*EXPERIMENT 2.* A more granular view can be obtained by focusing on the distribution of one decrypted symbol rather than all of $X^*$ This experiment examines the distribution of the first decrypted byte $p(X_1^*)$ in response to FLIP-D attacks against bits within the first ciphertext symbol. One subtle point is the following: if $Y_1^*$ happens to decrypt as an overflow, then $X_1^*$ actually corresponds to a decrypted version of $Y_2^*$, not of $Y_1^*$.

The plaintext (whose first byte is $X_1 = $ 0x20) was encrypted using PT[8,5,4] and the 24-bit NLFSR. For each encryption, one ciphertext bit (either $y_6$, $y_7$, $y_8$, or $y_9$) was flipped. Then, 4000 randomly selected keys were used, producing 1000 unique FLIP-D attacks against each of these four ciphertext bits. For each FLIP-D attack, the identity of decrypted symbol $X_1^*$ was tabulated. This process resulted in four empirical $p(X_1^*)$ distributions, which were plotted as 16 × 16 heat maps. Figure 9 shows this data, with panels A, B, C, and D representing $p(X_1^*)$ during attacks against bits #6, #7, #8, and #9, respectively.

These results hint at the difficulty of predicting the identity of the first decrypted byte even after changing just one bit of the first ciphertext symbol. For example, there is an obvious qualitative difference between Figure 9A and the other three distributions: the former is spread fairly evenly over most of its domain but has a single peak ($p(X_1^* = $ 0xA0$) \approx 0.32$), while the latter consist of four equiprobable values, which are assigned to different bytes in each case.

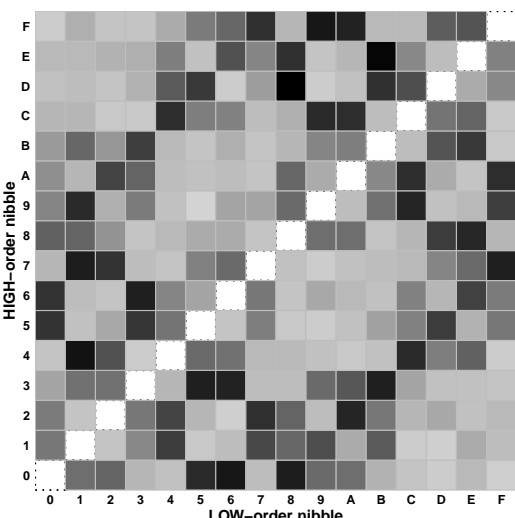

**Figure 8.** *Distribution of the decrypted bytes.* With PT[8,4,4], the 16-byte plaintexts `0x00`, `0x11`, `0x22`, and ... `0xFF` were encrypted 100 times, each with a unique secret key. Each ciphertext was then subjected to all possible FLIP-D attacks, and a histogram of the identity of decrypted bytes was produced. This histogram is illustrated as a heat map, with the high-order nibble of each byte on the Y-axis, the low-order nibble on the X-axis, and the byte frequency on a normalized gray scale with black representing $\sim$1.19%—the maximum observed frequency. Bytes corresponding to the original plaintext are set off along the diagonal from (0,0) to (0xF,0xF) as white squares surrounded by dotted lines (where white is for emphasis only and does not signify zero probability). Notice that, although the distribution does not appear uniform, every possible byte occurs in at least some decryptions.

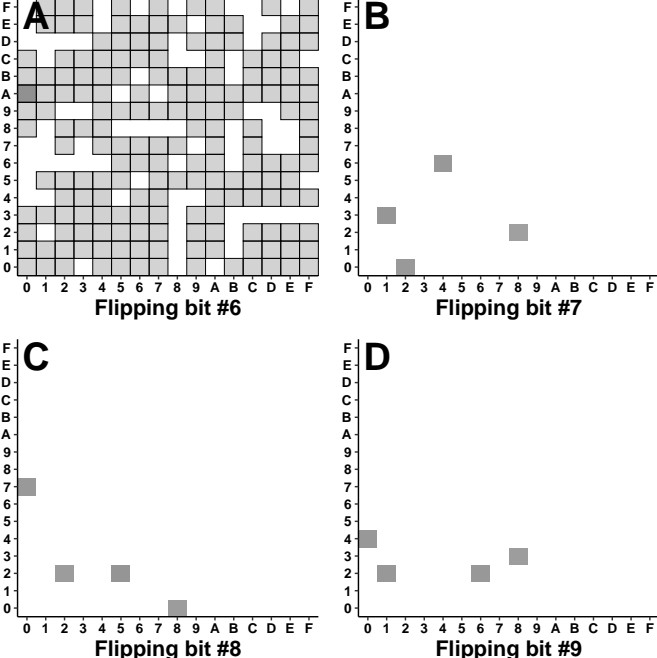

**Figure 9.** *Distribution of the first decrypted byte.* With PT[8,5,4], FLIP-D attacks were performed against each bit of the first discrepancy code ($y_6$, $y_7$, $y_8$, and $y_9$), with each attack being repeated with 1000 randomly chosen keys. Each panel shows the distribution of values taken by the first decrypted symbol, $p(X_1^*)$. Since this symbol is one byte long, the distribution can be represented as a $16 \times 16$ heat map with high-order nibbles on the Y-axis, low-order nibbles on the X-axis, and frequency as a grayscale. (**A**) During attacks against bit $y_6$, the first bit of the first discrepancy code, $X_1^*$ can take many different identities. (**B**) During attacks against $y_7$ however, $X_1^*$ only takes four different values—each with similar probability. Attacks against $y_8$ (**C**) and $y_9$ (**D**) also yield four nearly equiprobable bytes.

6.6.1. Attacks against Bit $y_6$: Qualitative Aspects

Here, we explain the qualitative (distribution shape) differences among these FLIP-D attack outcomes, starting with the attack against bit $y_6$. In this case, the first discrepancy code ($D_1^*$) was observed to take only four different values: $0000_2 = 0$, $0111_2 = 7$, $1000_2 = 8$, $1111_2 = 15$. At first, this seems at odds with Figure 9A. How can an attack produce only four discrepancy codes, and yet $X_1^*$ can still take so many ($\sim$180) different identities? The table below shows each of these new discrepancy codes being decrypted into $X_1^*$. Columns 1 and 2 are the original ($F_1 \parallel D_1$) and flipped ($F_1 \parallel D_1^*$) codewords. Column 3 gives the expression used during decryption to obtain $X_1^*$, and Column 4 evaluates this expression. Note that if $D_1 = 0111_2$ is changed to $D_1^* = 1111_2$, and the resulting codeword ($F_1 \parallel D_1^*$) will only be valid if $F_1$ happens to be $11111_2 = 31$—the 'all-1' failure counter associated with overflows (symbolized by ㉛ in the last row of Column 2).

| Original | Flip $y_6$ | First Decrypted Symbol ($X_1^*$) | |
| $F_1 \parallel D_1$ | $F_1 \parallel D_1^*$ | Expression | Value |
| --- | --- | --- | --- |
| $F_1 \parallel 1000_2$ | $F_1 \parallel 0000_2$ | $K_{F_1} \oplus 00000000_2$ | 0xA0 |
| $31 \parallel 1111_2$ | $31 \parallel 0111_2$ | $K_{31} \oplus 01000000_2$ | any byte |
| $F_1 \parallel 0000_2$ | $F_1 \parallel 1000_2$ | $K_{F_1} \oplus 10000000_2$ | 0xA0 |
| $F_1 \parallel 0111_2$ | ㉛ $\parallel 1111_2$ | $K_{F_2} \oplus D_2^*$ | any byte |

The most common decrypted symbol during this attack is $X_1^* = $ 0xA0. Row 1 shows how $X_1^*$ becomes 0xA0 when $D_1 = 1000_2 = 8$ (i.e., when $X_1$ matches keystream symbol $K_{F_1}$ everywhere except bit #8). In the absence of a bit-flipping attack, the first decrypted symbol 'should' therefore be $K_{F_1} \oplus 10000000_2$, which means that $K_{F_1}$ should be $X_1 \oplus$ 0x10000000$_2$ = 0x20 $\oplus$ 0x80 = 0xA0. A FLIP-D attack against $y_6$ changes the discrepancy code from $D_1 = 1000_2$ to $D_1^* = 0000_2$ while leaving failure counter $F_1$ unchanged. Thus, the first decrypted symbol in $Y^*$ becomes $X_1^* = K_{F_1} \oplus 00000000_2 = K_{F_1} = $ 0xA0.

Row 3 shows how $X_1^*$ becomes 0xA0 when $D_1 = 0000_2 = 0$ (i.e., when $X_1$ matches $K_{F_1}$ exactly—meaning that $K_{F_1} = $ 0x20). Here, the FLIP-D attack changes the discrepancy code from $D_1 = 0$ to $D_1^* = 1000_2 = 8$ without affecting the failure counter. Thus, when $Y^*$ is decrypted, its first symbol will be $K_{F_1} \oplus 10000000_2 = $ 0x20 $\oplus$ 0x80, which again equals 0xA0.

Besides 0xA0, the other values of $X_1^*$ appear to be scattered uniformly, as suggested by Rows 2 and 4 of the above table. In Row 2, an overflow occurs when encrypting the first plaintext symbol, and so decryption of the resulting 'all-1' codeword $(F_1 \parallel D_1) = (11111_2 \parallel 1111_2)$ would not produce any output. The FLIP-D attack, however, changes the discrepancy code from $1111_2$ to $0111_2$, without changing $F_1$. Since this new discrepancy code means 'a match everywhere except bit #7', decryption now produces $X_1^* = K_{31} \oplus 01000000_2$, which can take any value since, by definition, the keystream is pseudo-random.

Row 4 shows the other way in which $X_1^*$ can be decrypted into an arbitrary byte: the attack changes $D_1 = 0111_2$ into $D_1^* = 1111_2$. The original codeword ($F_1 \parallel D_1$) would have been interpreted as '$X_1$ matched $K_{F_1}$ everywhere except bit #7' and would have produced a decrypted symbol accordingly. The new codeword $(F_1 \parallel D_1^*) = (F_1 \parallel 1111_2)$ will only be valid if $F_1$ is also all 1's, in which case this new codeword will be interpreted as 'overflow'. Therefore, *no* decrypted symbol will be output, and the first symbol in the decrypted text ($X_1^*$) will actually be a decryption of the second ciphertext symbol $Y_2^*$ and not the first ciphertext-symbol. Since the second failure counter and discrepancy code are unrelated to $X_1$, the decryption can produce any value for $X_1^*$. (This assumes of course, that $Y_2^*$ does not also decrypt as an overflow. If it does, the same argument applies but with the third failure-counter and discrepancy-code rather than the second, and so on.)

6.6.2. Attacks against Bit $y_6$: Quantitative Aspects

This explains the qualitative appearance of $p(X_1^*)$ for the attack against bit $y_6$ (i.e., 0xA0 is relatively common but any byte is possible). Regarding the actual probabilities, 557

of these attacks were rejected and 443 led to successful decryptions. Among these, $X_1^*$ was observed to be `0xA0` 33.18% of the time (147/443) and to take all other values 66.82% of the time (296/443). To help explain these values, Columns 1 and 2 of the table below show the values of and observed frequencies (out of 1000) of the first discrepancy code, $D_1$; Columns 3 and 4 show the values and observed frequencies (out of 443) of the new code $D_1^*$ during the attack; and Column 5 gives the predicted frequencies (see below), assuming that $P_O / p_D$—the relative frequency of overflow to nonoverflow discrepancy-codes—remains fixed, where $P_O$ = 0.318144 from Equation (1).

| Original | | Flip $y_6$ Attack | | |
|---|---|---|---|---|
| $D_1$ | #/1000 | $D_1^*$ | #/443 | Predicted |
| $0000_2$ | 0.079 | $1000_2$ | 0.178330 | 0.161019 |
| $0001_2$ | 0.085 | - | 0 | - |
| $0010_2$ | 0.080 | - | 0 | - |
| $0011_2$ | 0.078 | - | 0 | - |
| $0100_2$ | 0.078 | - | 0 | - |
| $0101_2$ | 0.073 | - | 0 | - |
| $0110_2$ | 0.068 | - | 0 | - |
| $0111_2$ | 0.097 | $1111_2$ | 0.004515 | 0.001801 |
| $1000_2$ | 0.066 | $0000_2$ | 0.148984 | 0.161019 |
| $1111_2$ | 0.296 | $0111_2$ | 0.668172 | 0.676162 |

$D_1^* = 0000_2$ and $1000_2$ should occur with equal probability, which we denote as $p^*$. $D_1^* = 0111_2$ arises from the original discrepancy code for an overflow, $D_1 = 1111_2$, and thus has probability $(P_O / p_D) \cdot p^*$. Since the original discrepancy-code probabilities sum to unity, $(s + 1) p_D + P_O = 1$, and so $P_O / p_D = 4.199268$. Finally, $D_1^* = 1111_2$ can only be observed within an 'all-1' overflow codeword (i.e., when failure counter $F_1 = 11111_2 = 31$); otherwise, the attack would be rejected. Thus, $\Pr\{D_1^* = 1111_2\} = p^* \cdot \Pr\{F_1 = 31\} = (p^*) \cdot (1 - p)^{31} p$. Since the probability of these different $D_1^*$ values must sum to one,

$$\Pr\{D_1^* = 0000_2\} + \Pr\{D_1^* = 0111_2\} + \Pr\{D_1^* = 1000_2\} + \Pr\{D_1^* = 1111_2\} = 1$$

$$p^* + p^* \left( \frac{P_O}{p_D} \right) + p^* + p^* \Pr\{F_1 = 15\} = 1$$

from which we obtain $p^* = 0.161019$ and the other predicted probabilities in the last column of the table above.

6.6.3. Attacks on Bits $y_7$, $y_8$, and $y_9$

FLIP-D attacks against bits $y_7$, $y_8$, and $y_9$ behave similarly to each other but differently than from the attack against $y_6$. Rather than a large number of possible $X_1^*$ values, each of these other three FLIP-D attacks produces only four $X_1^*$ values. Interestingly, however, each of these three attacks can involve eight (not four) 'flipped' discrepancy codes. In the attack against $y_6$, a smaller number of discrepancy codes (4) lead to many $X_1^*$ values. Here, the opposite pattern is observed: a larger number of discrepancy codes (eight) lead to a smaller number (four) of $X_1^*$ values. Why does this occur?

Consider as an example the 1000 attacks against bit $y_9$, of which 638 succeeded and 362 were rejected. The observed frequencies of the original and 'flipped' discrepancy codes (along with predicted values for the latter) are shown below:

| Original | | Flip $y_9$ Attack | | |
|---|---|---|---|---|
| $D_1$ | #/1000 | $D_1^*$ | #/638 | Predicted |
| $0000_2$ | 0.079 | $0001_2$ | 0.133 | 1/8 |
| $0001_2$ | 0.085 | $0000_2$ | 0.123 | 1/8 |
| $0010_2$ | 0.080 | $0011_2$ | 0.122 | 1/8 |
| $0011_2$ | 0.078 | $0010_2$ | 0.125 | 1/8 |
| $0100_2$ | 0.078 | $0101_2$ | 0.114 | 1/8 |
| $0101_2$ | 0.073 | $0100_2$ | 0.122 | 1/8 |
| $0110_2$ | 0.068 | $0111_2$ | 0.152 | 1/8 |
| $0111_2$ | 0.097 | $0110_2$ | 0.107 | 1/8 |
| $1000_2$ | 0.066 | $1001_2$ | 0 | - |
| $1111_2$ | 0.296 | $1110_2$ | 0 | - |

$D_1$ = 0–8 each occur with probability $\sim p_D = (1 - P_O)/9 = 0.0758$. Although $1000_2$ and $1111_2$ are legal values for $D_1$, flipping the last bit of either will make it invalid (gray background). Thus, the 'flipped' discrepancy code $D_1^*$ can take eight possible values, each of which is observed to occur near its expected frequency of $\frac{1}{8}$.

The eight corresponding $D_1$ values can be paired into four 'dyads', in which one member becomes the other when $y_9$ is flipped, but both cause $Y_1^*$ to be decrypted into the *same* value. For example, discrepancy codes 4 and 5 are a dyad: flipping the final bit of 4 yields 5, and flipping the final bit of 5 yields 4. When the original discrepancy code is $D_1 = 5$, the first plaintext symbol and its matching the keystream symbol must have differed in their fifth bit. Thus, $K_{F_1} = X_1 \oplus 00010000_2 = 0x20 \oplus 0x10 = 0x30$. During the FLIP-D attack, however, $D_1 = 5$ becomes its dyadic partner $D_1^* = 4$. Under this new (incorrect) discrepancy code, $Y_1^*$ will be decrypted into $X_1^* = K_{F_1} \oplus 00001000_2 = 0x38$. Had the original discrepancy code been $D_1 = 4$ (i.e., the dyadic partner of $D_1 = 5$), then by the same reasoning $K_{F_1}$ would be $0x20 \oplus 00001000_2 = 0x28$. The bit-flip attack would alter this discrepancy code to $D_1^* = 5$, leading to the (incorrect) decryption $X_1^* = 0x28 \oplus 00010000_2 = 0x38$. Thus, both ($K_{F_1} = 0x30$, $D_1^* = 4$) and ($K_{F_1} = 0x20$, $D_1^* = 5$) produce the same decrypted symbol, $X_1^* = 0x38$.

In a similar manner, the other three dyads for the attack against $y_9$ (0 and 1, 2 and 3, 6 and 7) will produce the other three observed values of $X_1^*$. The probability of each value of $X_1^*$ is the sum of the probabilities of its corresponding dyad, which is just $\frac{2}{8} = 25\%$. (The same argument applies also for attacks against bits $y_7$ and $y_8$—only differing in the details of which discrepancy-codes are paired into which dyads.)

To summarize, we have shown how $p(X^*)$ can span the entire domain of 256 possible bytes, even though the original $X$ had only sixteen unique bytes. We have also discussed how flipping different bits within one discrepancy code (e.g., $y_6$ vs. $y_7$, $y_8$, or $y_9$) can produce qualitatively and quantitatively different distributions for the corresponding decrypted symbol.

*6.7. Reconnection*

One seemingly anomalous finding must still be explained. FLIP-F attacks should trigger an avalanche and therefore be rejected—especially for a lengthy plaintext like 800,000-bit message used here. However, Figure 7 shows that *some* (<1%) FLIP-F attacks are not. This is not due to mere statistical chance but rather due to a specific phenomenon called *reconnection*.

Recall that during normal PudgyTurtle decryption, $Y$ and $K$ are unlinked but connected. During a FLIP-D attack, $Y^*$ and $K$ usually stay connected: $K_F$ is still the appropriate keystream symbol to use for decryption (although the altered value of $D$ will cause $K_F$ to be reverse-engineered into the wrong plaintext symbol). During a FLIP-F attack, however, $Y^*$ and $K$ usually disconnect: $K_F$ is no longer the correct keystream symbol for reverse-engineering since by definition the failure counter $F$ is wrong. Disconnection then causes each subsequent ciphertext symbol to be 'unmasked' into an effectively random value rather than the appropriate codeword, producing an arbitrary decryption with a high

probability of being rejected. If rejection does not occur, it is likely that $K$ and $Y^*$ have re-connected. This occurs by chance if the *running total* of keystream symbols consumed while decrypting $Y^*$ up to some point coincides with the running total of keystream symbols that would have been consumed decrypting $Y$ up to that same point.

To illustrate reconnection, consider an example using PT[8,5,4] and a randomly keyed 24-bit NLFSR, in which 1000 FLIP-F attacks are mounted against bit #1 (the first failure-counter bit). To better visualize the raw data, the plaintext is a 32768-bit message containing a simple repeating 16-byte test pattern. As expected, most attacks were rejected (990/1000), but a few (0.1%) led to successful decryptions. The partial hex dump of one such decryption (secret key `0x5A286F`) is shown below, along with the original plaintext:

| | |
|---|---|
| $X$ (original) | `00112233 44556677 8899AABB CCDDEEFF 00112233...` |
| $X^*$ (BFA) | `A4480333 44556677 8899AABB CCDDEEFF 00112233...` |

The first three decrypted bytes differ from the original plaintext, but all remaining bytes are identical. The cartoon below compares the failure counters during decryption of the original and 'flipped' ciphertexts. Column 1 shows the index of the symbol being decrypted. Failure counters are shown for decryption of each symbol in $Y$ (Column 2) and $Y^*$ (Column 4). Running totals ($\sum$) of the required number of keystream symbols for plaintext-to-keystream matching which are given for $Y$ (Column 3) and $Y^*$ (Column 5). (Recall that for failure counter $0 \leq F < 2^f$, an overflow or successful plaintext-to-keystream match consumes $F + 1$ keystream symbols). The running-total does not include the keystream involved in constructing each mask. Including this data would not change the results but would require keeping a running total of $s(F+1) + c$ keystream *bits* for each entry, rather than of $F + 1$ keystream *symbols*.)

| | **Reconnection during a FLIP-F Attack** | | | |
|---|---|---|---|---|
| | **Original ($Y$)** | | **Flipped ($Y^*$)** | |
| | $F$ | $\sum$ | $F$ | $\sum$ |
| 1 | 13 | 14 | 29 | 30 |
| 2 | 23 | 38 | 14 | 45 |
| 3 | 25 | (64) | 18 | (64) |
| 4 | 21 | 86 | 21 | 86 |
| 5 | 21 | 108 | 21 | 108 |
| ⋮ | | ⋮ | | ⋮ |

The running-totals match after decryption of the third symbol ($\sum = 64$, circled). Thus, $X_1^*$, $X_2^*$ and $X_3^*$ may differ from $X_1$, $X_2$, and $X_3$, but will be the same thereafter. During this FLIP-F attack, $Y^*$ disconnects from $K$ immediately (during decryption of $Y_1^*$), but then—due to a coincidence of running-totals—reconnects after decryption of the third ciphertext symbol. Decrypting either $Y$ or $Y^*$ up to this point would require 64 keystream symbols.

It would seem that the attacker could leverage this behavior to their advantage by tailoring a BFA to affect only a certain segment of plaintext, but reconnection is an uncertain prospect. Although it is easy to visualize after the fact (given $Y$, $Y^*$, and $K$), it is difficult to know in advance whether or not it will occur, and—if it does—where exactly it will happen. Moreover, BFAs in this paper are limited to flipping only one bit. For attacks involving multiple bit flipping, the chance of maintaining a reconnected state for the entire remaining $Y^*$ would diminish. Both of these factors make it harder for an attacker to exploit a reconnection (also see Section 7).

### 6.8. Symbol Insertion

Bit-flipping attacks against PudgyTurtle can also produce an unusual behavior called *symbol insertion*, in which the decryption contains an 'extra' symbol. This violation of length

preservation appears to be unique to PudgyTurtle and would not occur during bit-flipping attacks against other stream-cipher encryption modes.

The root cause of symbol insertion is overflows: when a flipped bit changes an overflow symbol into a valid nonoverflow codeword, a new decrypted symbol is produced rather than the 'no output' that would otherwise have occurred. Flipping bit $y_6$ while using PT[4,5,3], for instance, might change an overflow codeword ($11111111_2$) into $11111011_2$. The former would not produce any decrypted symbol, but the latter—which PudgyTurtle would interpret as "$K_{31}$ matched the plaintext-symbol everywhere except bit #3"—would produce the 'extra' decrypted symbol $K_{31} \oplus 0100_2$.

Data from Section 6.7 also provide an example of symbol insertion. Hex dumps from this attack (with secret key 0x9D5EA3) are as follows:

| | |
|---|---|
| *X* (original) | 00112233 44556677 8899AABB CCDDEEFF 00112233... |
| *X*\* (BFA) | D7201722 33445566 778899AA BBCCDDEE FF001122... |

After its third byte, $X^*$ is immediately recognizable as $X$ shifted rightward by one byte. The failure counters and running totals for this attack are presented below.

| | **Symbol Insertion during a FLIP-F Attack** | | | | |
|---|---|---|---|---|---|
| | **Original (*Y*)** | | | **Flipped (*Y*\*)** | |
| | *F* | $\Sigma$ | | *F* | $\Sigma$ |
| 1 | 15 | 16 | | 31 | 32 |
| 2 | (31) | 48 | | 24 | 57 |
| 3 | 17 | ⑥⑥ | | 8 | ⑥⑥ |
| 4 | 16 | 83 | | 16 | 83 |
| 5 | 19 | 103 | | 19 | 103 |
| ⋮ | | ⋮ | | | ⋮ |

$Y_2$ is normally decrypted as an overflow—indicated by parenthesis around its associated failure counter ($F$ = 31 in Row 2)—and therefore has no corresponding output symbol. Thus, $Y_1$ and $Y_2$ together only yield *one* decrypted symbol. On the other hand, during the bit-flip attack, $Y_2^*$ no longer decrypts as an overflow, and so $Y_1^*$ and $Y_2^*$ together produce *two* decrypted symbols: $X^*$ now contains one more symbol than did the original plaintext (To be clear, the first failure counter during the bit-flip attack ($F_1^*$ = 31, Row 1, Column 4) is the same value that occurs during an overflow. However, the discrepancy code paired with $F_1^*$ is not $1111_2$, and so the resulting codeword of ($F_1^* \parallel D_1$) is not interpreted as an overflow but rather as a plaintext-to-keystream match between $X_1$ and $K_{31}$. This numerical coincidence has nothing to do with reconnection or symbol insertion). Reconnection occurs after the third ciphertext symbol (identical running-totals ⑥⑥), and thus the remainder of $Y^*$ is decrypted correctly, but each symbol's *position* is shifted one *s*-bit (8-bit) symbol rightward compared to $X$.

Illustrated differently, the decryption of the original (unmodified) ciphertext—including the $Y_2$-overflow—is as follows:

$$
\begin{array}{cccccc}
Y_1 & Y_2 & Y_3 & Y_4 & Y_5 & \ldots \\
\downarrow & \downarrow & \downarrow & \downarrow & \downarrow & \\
X_1 & & X_2 & X_3 & X_4 & \ldots \\
\text{0x00} & & \text{0x11} & \text{0x22} & \text{0x33} & \ldots
\end{array}
$$

Meanwhile, the decryption of the flipped ciphertext is as follows:

$$
\begin{array}{cccccc}
Y_1^* & Y_2^* & Y_3^* & Y_4^* & Y_5^* & \ldots \\
\downarrow & \downarrow & \downarrow & \downarrow & \downarrow & \\
X_1^* & X_2^* & X_3^* & X_4^* & X_5^* & \ldots \\
\text{0xD7} & \text{0x20} & \text{0x17} & \text{0x22} & \text{0x33} & \ldots
\end{array}
$$

$Y_4^*$ correctly decrypts to 0x22, for example, but this byte is now the *fourth* decrypted symbol, not the third.

Another clue about symbol-insertion comes from the normalized Hamming distance. For this attack, $\overline{\mathrm{Hd}}$ = 46.9%. This value seems odd: although $\overline{\mathrm{Hd}}$ is nearly 50%, the decryption is not 'random-looking' at all but quite recognizable as a shifted version of the original test pattern.

When comparing a message to a shifted version of itself, the normalized Hamming distance may take a recognizable value. With natural languages, this value is known (e.g., for a one-byte shifted version of our ASCII-encoded English plaintext source, it would be $\approx 36.2\%$), and for study purposes, it could even be manipulated with specially designed plaintexts (e.g., $\overline{\mathrm{Hd}}$ would be '1' for the plaintext 0xFF00FF00... shifted by one byte, etc.). For a 1-byte shift and this particular test pattern, $\overline{\mathrm{Hd}}$ just happens to be nearly 50%—the same value expected during an avalanche.

The original intuition about FLIP-F attacks triggering avalanches, altering many bits of $Y^*$ and being rejected, is sound. However, among the few FLIP-F attacks that are not rejected, reconnection and sometimes symbol insertion will be observed. A rejected ciphertext should have $\overline{\mathrm{Hd}} \approx 50\%$, but in practice, there may not be any data to analyze. With reconnection, $\overline{\mathrm{Hd}}$ reflects the number of bits over which disconnection persists. With symbol insertion, $\overline{\mathrm{Hd}}$ may take a recognizable value for a certain plaintext (or class of plaintexts).

## 7. Robbing the Bank

This section discusses a bit-flip attack against a hypothetical banking transaction. The goal of this exercise is to change a very simple 15-byte plaintext ("DEPOSIT:$500.00") into a new message specifying a larger deposit. The decryption must match the original format except that it may contain up to 4 digits before the decimal point—thus allowing the attacker to take advantage of potentially inserting an 'extra' decrypted symbol to deposit more than $999.99.

To preview the results, most attacks fail. Among successful attacks, only a few decryptions are 'meaningful': most violate pre-specified formatting guidelines. Among meaningful decryptions, the deposited amount is unpredictable: it may be less than, greater than, or exactly equal to $500. Among successful deposits >$500, the profit varies over a ten-fold range. Although PudgyTurtle does not completely prevent successful attacks, it adds significant uncertainty about which bit should be flipped to make a profit.

This experiment uses PT[8,4,4] and RC4 with 1000 randomly chosen 40-bit secret keys. For each key, attacks were performed against ciphertext symbols starting with $Y_{10}$ (the first one that might represent an encrypted digit—assuming no overflows) and ending with the penultimate ciphertext symbol, $Y_{N_Y-1}$. For each ciphertext-symbol, all eight of its bits were flipped in succession, resulting in 223,272 individual bit-flip attacks in total. Each attack was categorized into one of four outcomes:

- **REJECT:** decryption returned $\perp$ due to an invalid discrepancy-code.
- **NONSENSE:** decryption occurred, but was not meaningful. The bank itself (not the decryption algorithm) would likely reject these transactions for being ill-formatted. Meaningful decryptions were required to contain the string "DEPOSIT:$" followed by 1–4 decimal digits, a decimal point, and two more digits. Nonsense decryptions included things like "DEPzSIT:$100.00" (mis-spelling), "DEPOSIT:$50a.00" (nondigital value), "DEPOSIT:$500+00" (no decimal point), "DEPOSIT:$.05" (too few digits before the decimal point), and "DEPOSIT:$500.1" (too few digits after the decimal point). Leading zeros (e.g., "DEPOSIT:$0010.00") and null transactions (e.g., "DEPOSIT:$000.00"), however, were accepted.
- **LOSS:** a meaningful decryption specified a deposit $\leq\$500.00$. A $500.00 deposit exactly also counted as a 'LOSS' due to the uncompensated time and effort required to mount the attack
- **GAIN:** a meaningful decryption produced a deposit >$500.00.

Figure 10A shows that valid decryptions were rare: about three-quarters of BFAs were rejected (76.1%), and most others decrypted into nonsense (21.4%). *Meaningful* decryptions—by which we mean those that were not rejected and decrypted into something besides nonsense—occurred only ≈2.5% (5604/223272) of the time. The ultimate goal of this exercise (GAIN) was even less common—although gains did constitute a majority of meaningful decryptions (4667/5604 = 87%, but only 4667/223272 = 2.1% of all attacks).

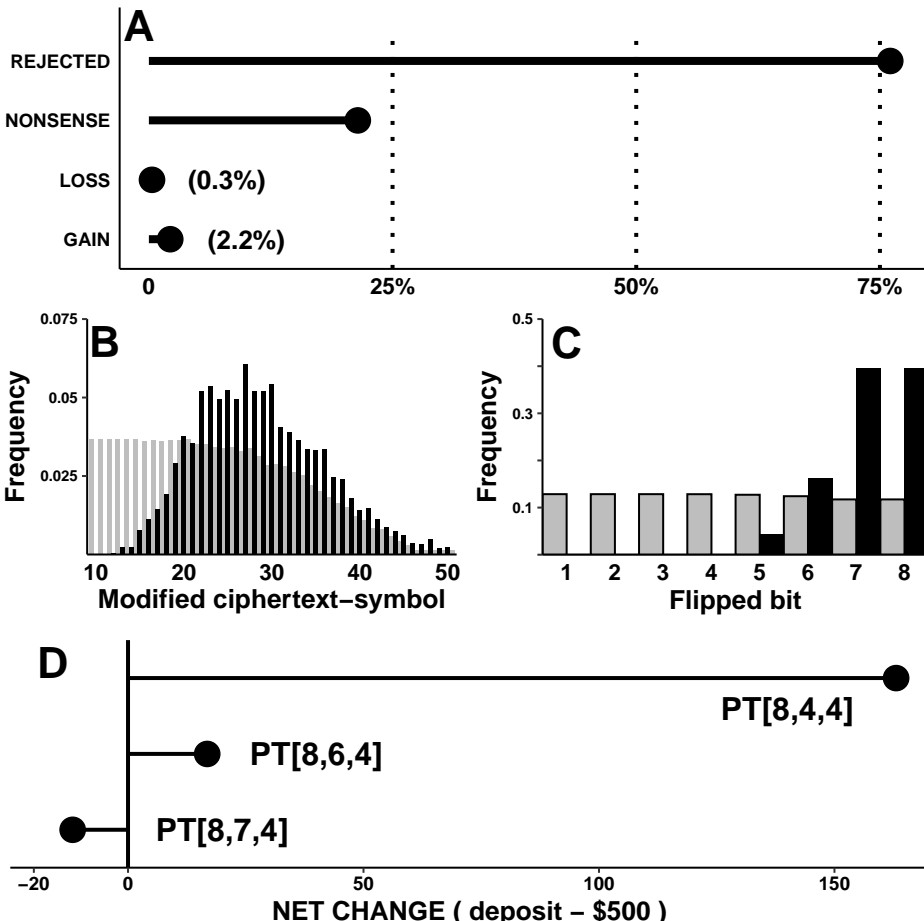

**Figure 10.** *Attacking a hypothetical bank transaction.* Over 220,000 bit-flipping attacks were mounted against a hypothetical PudgyTurtle-mode-encrypted message specifying a $500 deposit. (**A**) This histogram of attack outcomes shows that most attacks were either rejected (REJECTED) or produced incorrectly formatted output (NONSENSE). Only ∼2.5% led to 'meaningful' decryptions, categorized as LOSS or GAIN. (**B**) This histogram illustrates which ciphertext symbol contained the flipped bit for attacks with meaningful decryptions (■ bars) and for attacks in the rejected/nonsense category (■ bars). (**C**) A histogram showing which bit (within any 8-bit ciphertext symbol) was flipped, again representing meaningful decryptions as ■ and rejected/nonsense outcomes as ■. (**D**) The average net change (i.e., 'decrypted deposit' minus $500) for successful attacks against PudgyTurtle implementations PT[8,4,4], PT[8,6,4], and PT[8,7,4]. Depending on the implementation (and specifically, its failure counter $f$ = 4, 6, or 7), attacks could produce more profit ($163), less profit ($16), or even a loss (−$11).

Can the attacker predict which bit to target in order to have the bank accept the transaction? The results suggest not. Figure 10B shows histograms of which ciphertext symbol contained the flipped bit, both for meaningful decryptions (black bars, ■) and for rejected attacks or nonsense decryptions (gray bars, ■). Although these two histograms have different shapes, they overlap significantly. Thus, targeting symbols that occur early or late in the ciphertext is suboptimal: these symbols coincide with low-probability tails of

the black ('meaningful') histogram. Yet, targeting the remaining ciphertext symbols near the peak of the black curve is also not ideal: their positions fall along the plateau of the gray ('nonsense/rejected') histogram. Therefore, such attacks are equally likely to fail as attacks against early ciphertext symbols.

Figure 10C shows similar information about which bit (within any 8-bit ciphertext-symbol) was flipped. While FLIP-D attacks against the two final bits of any codeword seems like the best strategy, such attacks still have a similar probability of REJECTED/NONSENSE outcomes as do attacks against the other six bits.

So far, GAIN and LOSS have been lumped together as 'meaningful' decryptions. However, it is not clear how certain the attacker be about the actual dollar amount of the final result or which bit should be flipped to turn a profit. After all, decrypted deposits varied widely—from $0.00 to $9500.00. If we define the net change as a 'decrypted deposit' minus $500.00, then its average would $163.12—reflecting an average LOSS of $-$272.25 and an average GAIN of $227.00.

Higher profits occur among decryptions with an extra symbol (i.e., changing the dollar amount before the decimal point from a 3-digit to a 4-digit value). Since symbol insertion occurs when an overflow codeword is changed into something else, extra digits should be more common when overflows are more common, and vice-versa. The attack so far has only involved a single PudgyTurtle implementation (PT[8,4,4]) with a substantial overflow rate of $P_O = 56.40\%$. Would 'robbing the bank' be less profitable in situations with fewer overflows?

To test this, another set of bank robberies was attempted against implementations with larger failure counters ($f = 6$ and 7) and therefore fewer overflows (i.e., $P_O = 10.12\%$ for PT[8,6,4] and 1.02% for PT[8,7,4]). The rest of the protocol was identical to the one above, but this time only 100 keys were used. For each implementation, the net change was averaged over all meaningful decryptions. As illustrated in Figure 10D, overflows indeed affected the profit margin. When overflows were most likely (PT[8,4,4]), the average net change was a GAIN of $163; when overflows were somewhat less likely (PT[8,6,4]), the net-change was a GAIN of only $16; and when overflows were rare (PT[8,7,4]), the net change was a LOSS of $11.

This analysis adds little to what the bank robber could simply have assumed from the outset: the best strategy is a FLIP-D attack against some ciphertext symbol 'in the middle' that probably represents an encrypted plaintext digit. While these observations can be taken as limited conclusions for a 1-bit attack against one particular plaintext and PudgyTurtle implementation, a broader interpretation is also possible: predicting the actual results of bit-flipping attacks in the presence of PudgyTurtle is difficult. The ultimate outcome depends on factors controlled by the attacker (i.e., which bits(s) are flipped) and by the sender/receiver (i.e., implementation parameters $s$, $f$, and $d$) but also has some inherent uncertainty due to the stochastic nature of PudgyTurtle mode's plaintext-to-keystream matching process.

## 8. Discussion

This manuscript has explored how the stream-cipher encryption mode PudgyTurtle affects ciphertext malleability in the context of bit-flipping attacks. In these attacks, the opponent captures the ciphertext, alters one or more bits, and then retransmits this modified ciphertext. Here, we investigated only the simplest attack: flipping a single bit.

PudgyTurtle's plaintext-to-keystream matching process implements a nonsystematic, 1-bit error-correcting code, transforming an $s$ bit input into a $c = f + d$ bit output. Many other stream-ciphers also utilize error-correcting codes, such as the 'noisy keystream encryption' (NKE) model of Kara and Erguler [25,26] and several variants of the 'learning parity with noise' (LPN) problem [27] including LPN-C by Gilbert and colleagues [28], a matrix-based system by Applebaum [29], and a homophonic system by Mihaljevic and Imai [30]. In these examples, the plaintext is typically encoded with an error-correcting code, and then the codewords are purposely mixed with noise as well as keystream. This 'encode

before enciphering' idea has been analyzed in depth by Bellare and Rogaway [31] and even relates in some ways back to Shannon's observations about how easy it would be to encipher a perfectly- ncoded (i.e., zero redundancy) artificial language [32]. What differentiates PudgyTurtle mode from these other systems is that the underlying motivation for the ECC is not to combat *external* noise but rather to describe the match (to within a 1-bit tolerance) between each plaintext symbol and some keystream symbol. Essentially, the keystream functions as an 'internal' noise source, against which each plaintext symbol is compared. The stochastic nature of each plaintext-to-keystream match introduces history (feedback) into the encryption process, thereby adding qualitative and quantitative uncertainty to the outcome of bit-flipping attacks.

Good protection against bit-flipping can be provided by authenticated encryption modes (Chapter 4 [33]) or message-authentication codes (Section 18.14 [34]). However, these authentication strategies themselves may be vulnerable. The MAC, for instance, has security issues including attacks based on replay, length-extension, padding, variable key lengths, collisions of suffix-based constructions, and side channels (timing of string comparisons) [35–37], (Chapter 7 [38]), (Chapter 4 [12]), (Chapter 3 [33]). Therefore these methods may necessitate yet another layer of complexity (e.g., sharing another [authentication] key, including a counter or nonce, lengthening the authentication tag size above some threshold, and so on).

Another protection against bit-flipping is to use a stream-cipher encryption mode other than synchronous (S-BASC). These modes enhance message integrity to some degree, and thus offer limited defense against bit-flipping—although not the full protection afforded by a MAC. Taking the 'effect' of a one-bit BFA to mean the point at which $X^*$ begins to diverge from $X$ and the number of bits over which this continues, we can say, for example,that against S-BASC mode, the effect is immediate and short lived (one bit). Against the asynchronous/self-synchronizing mode, the effect is also immediate, but persists for $n$ bits (the size of the KSG inner state). Against PudgyTurtle mode, the effect has a variable starting point *and* duration. This unpredictability is because PudgyTurtle includes uncertainty about how much keystream will be required to encrypt each plaintext symbol.

This paper first introduced a 'generalized' PudgyTurtle implementation (PT[$s, f, d$]) which allows variably sized input and output symbols. This implementation takes $s$-bit plaintext symbols as input and produces $c$-bit ciphertext symbols as output, where $c = f + d$, $f$ is the failure-counter size, and $d$ the discrepancy-code size. In previous research using one particular implementation (PT[4,5,3]), overflows were rare, its bandwidth expansion was $\sim$2, and its keystream expansion was $\sim$5.2. For generalized PudgyTurtle, overflows can be more common (even occurring as multioverflow events) and scales with $s/f$, bandwidth expansion scales with $c/s$, and keystream expansion with $s$. Many ($s, f, d$) combinations can achieve both adequate speed and compactness (i.e., keystream expansion <5–10, and bandwidth expansion <2–3), with Equations (1), (3), and (4) predicting the performance of any proposed PT[$s, f, d$] implementation.

Next, this paper explored the several ways in which PudgyTurtle mode affects message integrity during single-bit bit-flipping attacks:

- The attacker may be able to predict the effects of a BFA (i.e., the first and last affected bits) on average, but not exactly.
- The decryption algorithm itself rejects attacks that produce an invalid codeword. Each codeword is the concatenation of an $f$-bit failure counter and a $d$-bit discrepancy code ($F \parallel D$). In PudgyTurtle, all $2^f$ failure counters are possible, but only ($s + 2$) discrepancy codes are needed—not all $2^d$ of them. A BFA may produce one of these unassigned discrepancy codes or may lead to an incorrect pairing of the 'all-1' discrepancy code (associated with overflows) with an $F$ that is *not* all 1's. Either way, the resulting codeword will be invalid.
- Depending upon which bit within a $c$-bit ciphertext-symbol is flipped, attacks can be grouped into those that alter a failure counter (FLIP-F) and those that alter a discrepancy code (FLIP-D). Rejections are more likely during FLIP-F attacks than FLIP-D

attacks. When a failure counter is altered, the keystream and ciphertext become disconnected, after which the decryption algorithm returns seemingly random outcomes. When a discrepancy-code is altered, in contrast, the keystream and ciphertext remain connected: except for one affected ciphertext-symbol, decryption proceeds normally, affecting only a few bits overall.

- Flipping even a single ciphertext-bit can change the distribution of decrypted symbols substantially compared to the original plaintext, altering its domain (all vs. just a few *s*-bit symbols) and shape (uniform vs. peaked). These distributions can be computed on a case-by-case basis but may not be amenable to a generalized formula.
- Avalanches sometimes self-terminate. If the keystream and ciphertext 'reconnect', the attack will not be rejected. Reconnection happens when the running-total number of keystream symbols used to decrypt the modified ciphertext coincidentally equals what it would have been for the original ciphertext. The length of disconnected ciphertext and the position at which reconnection happens are both uncertain, however, due to the statistical nature of each plaintext-to-keystream match.
- A new symbol is sometimes inserted into the decryption during a BFA. This occurs when a ciphertext symbol that should have been unmasked as an overflow (and therefore not trigger any output) is instead unmasked as a codeword that specifies a plaintext-to-keystream match (and therefore does trigger the output of a decrypted symbol). Symbol insertion also involves uncertainty with respect to its location and identity.

Together, these effects of PudgyTurtle mode make it harder for an attacker to tailor the correct set of bit flips to achieve a desired outcome and increase the likelihood that the modified ciphertext will be rejected. In the final section of this paper, we attacked a hypothetical bank transaction with the goal of depositing more than the specified amount of money. The results of this experiment confirmed all of these observations: most attacks were rejected by the decryption algorithm itself; most attacks that were decrypted contained obvious formatting errors which would not have been accepted by the bank; the content of the remaining decryptions ($\sim$2% of all attacks) was uncertain: losses could occur as well as gains, and even among attacks resulting in a net gain, the actual profit could vary substantially.

One limitation of this manuscript is only including BFAs in which a single bit was flipped. This simple case made it easier to interpret the effects of bit-position on attack outcomes but artificially limited the power of each attack. However, some of the observed effects may be even less likely if multiple bit flips are allowed. Reconnections, for example would likely be shorter if multiple bit-flips caused disconnection at many points. Another limitation is the toy ciphers used in our experiments: a 40-bit RC4 and a simple 24-stage NLFSR. While neither of these is appropriate for secure systems, both can nevertheless generate sufficiently random keystream for the purposes of this study. Potentially, however, subtle bit correlations could have influenced some of our results (e.g., attacks targeting $Y_1$, the outcomes of which rely more heavily on early keystream). To reduce this effect, we chose KSG starting states at random and when feasible used a different one for each bit-flip attack.

The extent to which various stream-cipher modes increase the work effort and decrease the success-rate of bit-flipping attacks remains an open question. PudgyTurtle mode by itself should not be relied upon to prevent such attacks. Future research, however, may focus on improving its effectiveness in this regard. For example, as probabilistic encryption (such as the 'noisy keystream' [25,26]) could in theory render tailored bit-flipping infeasible, could PudgyTurtle also be modified to achieve randomized encryption?

**Author Contributions:** Conceptualization, D.A.A. and A.C.S.; methodology, D.A.A. and A.C.S.; software, D.A.A.; validation, D.A.A.; formal analysis, A.C.S. and D.A.A.; writing—original draft preparation, D.A.A.; writing—review and editing, A.C.S. and D.A.A. All authors have read and agreed to the published version of the manuscript.

**Funding:** This research received no external funding.

**Data Availability Statement:** Software for the PT[4,5,3] PudgyTurtle instance is available at https:// github.com/smaugust/PudgyTurtle. Software for generalized PT[$s, f, d$] instances is being developed for GitHub release as well.

**Conflicts of Interest:** The authors declare no conflict of interest.

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
