# Peer review of "PudgyTurtle Mode Resists Bit-Flipping Attacks"

_cryptography, doi:10.3390/cryptography7020025_

Round 1

Reviewer 1 Report

The authors published PudgeTurtle and its effects on resisting TMDTO attacks in 2020. This paper is a further research for Pudgeturtle, which concentrate on the one-bit bit-flipping attacks. This paper is clear and well-structured. Sufficient experimental designs and results are provided to evaluate the bit-flipping attack resistance ability. Besides, a generalized PudgeTurtle allowing for variably-sized symbols is proposed in this paper.

Further text editing error checks are required. For example, there are two ”and” in Line 52.

Author Response

See attached PDF letter

Reviewer 2 Report

The manuscript was presented in a proper format. The authors included the proposal about the PudgyTurtle mode among the other modes in a nice concept. The flow of the paper may have a small rate of complexity, which is difficult to follow some concept. However, I would recommend this manuscript for publication because it provides a great contribution in the cryptographic system.  

Below are the comments:

1. What is the main question addressed by the research?

Encoding a plaintext can be done using many cryptographic techniques and methods based on the received bits as a stream or blocks. This paper discusses a stream-mode in which the main issue (attack) during bit exchange is bit flipping (injection). This paper proposed a mode that would prevent bit-flipping.

2. Do you consider the topic original or relevant in the field? Does it address a specific gap in the field?

I think the idea of bit-flipping has been a huge concern for cryptographers for awhile, but this proposal addresses a part of it. Definitely, it's considered a proposal for a solution, where not necessary all the proposals are valid.

3. What does it add to the subject area compared with other published material?

I think I answered this in the first question. Any idea is worth having a place for discussion.

4. What specific improvements should the authors consider regarding the methodology? What further controls should be considered?

I like the heavy information provided. However, my comment is only meant to fix the flow of the concepts presented in the manuscript.

5. Are the conclusions consistent with the evidence and arguments presented, and do they address the main question posed?

Yes, the evidence and discussion are provided.

6. Are the references appropriate?

No comments on the references. I think they are fine.

7. Please include any additional comments on the tables and figures.

Figures and tables can deliver the idea. However, they can be better, but they are reasonable and understandable.

Author Response

See attached PDF / letter

Reviewer 3 Report

This paper focus on bit-flipping attack for Pudgy Turtle mode for stream-cipher system. This paper shows that only partial protection against bit-fippling attack is given by Pudgy Turtle mode. I think it is very useful result. From the perspetive of security, since other method like MAC can give more efficient protection against bit-flipping attack, I think using Pudgy Turtle mode for stream-cipher system to resist bit-flipping attack is not a good idea. 

Author Response

See attached PDF / letter

Reviewer 4 Report

This paper analyzes bit-flipping attacks on the relatively new PudgyTurtle encryption mode.  The descriptions, analysis, and conclusions are well done.  I believe the paper is very close to acceptable in its current form. 

A few suggestions for improvement:

1) It appears that the authors only recently introduced PudgyTurtle mode in a previous paper.  Has anyone else acknowledged and cited that work?  Has any work on PudgyTurtle been done by anyone other than the authors.  If so, this would strengthen the paper as it would show that other professionals are interested in the results.

2) Regarding the "heat maps," it would help if the authors explain the meaning of the dark and light boxes. 

3) Minor spell check/grammar check can catch a few typos.

Author Response

See attached PDF / letter
